# The oxidative stress response, in particular the *katY* gene, is temperature-regulated in *Yersinia pseudotuberculosis*

**Daniel Scheller**[1]**, Franziska Becker**[1]**, Andrea Wimbert**[1]**, Dominik Meggers**[1]**, Stephan Pienkoß**[1]**, Christian Twittenhoff**[1]**, Lisa R. Knoke**[2]**, Lars I. Leichert**[2]**, Franz Narberhaus**[1]*

**1** Ruhr University Bochum, Faculty of Biology and Biotechnology, Microbial Biology, Bochum, Germany,
**2** Ruhr University Bochum, Faculty of Medicine, Institute of Biochemistry and Pathobiochemistry, Microbial Biochemistry, Bochum, Germany

\* franz.narberhaus@rub.de

**Data Availability Statement:** All relevant data are within the paper and its Supporting Information files.

## Abstract

Pathogenic bacteria, such as *Yersinia pseudotuberculosis* encounter reactive oxygen species (ROS) as one of the first lines of defense in the mammalian host. In return, the bacteria react by mounting an oxidative stress response. Previous global RNA structure probing studies provided evidence for temperature-modulated RNA structures in the 5'-untranslated region (5'-UTR) of various oxidative stress response transcripts, suggesting that opening of these RNA thermometer (RNAT) structures at host-body temperature relieves translational repression. Here, we systematically analyzed the transcriptional and translational regulation of ROS defense genes by RNA-sequencing, qRT-PCR, translational reporter gene fusions, enzymatic RNA structure probing and toeprinting assays. Transcription of four ROS defense genes was upregulated at 37°C. The *trxA* gene is transcribed into two mRNA isoforms, of which the most abundant short one contains a functional RNAT. Biochemical assays validated temperature-responsive RNAT-like structures in the 5'-UTRs of *sodB*, *sodC* and *katA*. However, they barely conferred translational repression in *Y. pseudotuberculosis* at 25°C suggesting partially open structures available to the ribosome in the living cell. Around the translation initiation region of *katY* we discovered a novel, highly efficient RNAT that was primarily responsible for massive induction of KatY at 37°C. By phenotypic characterization of catalase mutants and through fluorometric real-time measurements of the redox-sensitive roGFP2-Orp1 reporter in these strains, we revealed KatA as the primary $H_2O_2$ scavenger. Consistent with the upregulation of *katY*, we observed an improved protection of *Y. pseudotuberculosis* at 37°C. Our findings suggest a multilayered regulation of the oxidative stress response in *Yersinia* and an important role of RNAT-controlled *katY* expression at host body temperature.

**Funding:** Funding was provided by the German Research Foundation (DFG NA 240/10-2 and NA 240/14-1 to FN, LE 2905/1-2 to LIL, and Research Training Group 2341 "Microbial Substrate Conversion (MiCon)" to FN and LIL). The funders had no role in study design, data collection and analysis, decision to publish, or preparation of the manuscript.

**Competing interests:** No competing interests.

## Author summary

The external conditions dramatically change when a bacterial pathogen enters a mammalian host. Sensing the new situation and rapidly responding to it is of critical importance for pathogens, like *Yersinia pseudotuberculosis*, since they often circulate between their environmental reservoirs and a warm-blooded host. Many virulence-related genes encode a temperature-sensitive mRNA element, a so-called RNA thermometer (RNAT), in the 5'-end of their transcript. Melting of this structure at 37°C allows ribosome binding and translation initiation. The host immune system typically fights microbial pathogens by the production of reactive oxygen species (ROS). Here, we find that several ROS defense genes in *Yersinia* are upregulated at host body temperature to counteract the ROS attack. In particular, the massive RNAT-mediated upregulation of the catalase KatY confers protection against $H_2O_2$ at 37°C. Our study reveals a close regulatory link between temperature sensing and the oxidative stress response in a notorious food borne pathogen.

## 1 Introduction

All bacteria encounter oxidative stress caused by reactive oxygen species (ROS), such as superoxide ($O_2^-$), hydrogen peroxide ($H_2O_2$) and hydroxyl radicals [1]. ROS either accumulate endogenously by processes like aerobic metabolism and iron-sulfur cluster oxidation [2] or by exogenous exposure through their environment. Environmental $H_2O_2$ concentrations can rise due to excretion of $H_2O_2$ by lactic acid bacteria or as a response of the host immune response [3]. These sources of exogenous ROS are especially relevant for pathogenic bacteria, causing intestinal infections during gut colonization [4]. One such bacterium is the food borne pathogen *Yersinia pseudotuberculosis*, which leads to gastrointestinal diseases, like acute abdominal pain, mesenteric lymphadenitis or diarrhea [5].

To overcome oxidative stress induced challenges, bacteria have developed multiple strategies to combat ROS. $O_2^-$ can be transformed by superoxide dismutases into $O_2$ and $H_2O_2$, which can be further neutralized by catalases into $H_2O$ and $O_2$. Furthermore, $H_2O_2$ is scavenged by alkyl hydroperoxide reductases [3]. ROS often oxidize a variety of proteins containing cysteine residues. This results in non-native disulfide bond formation, which often causes a loss of function. Antioxidants, such as thioredoxins and glutathione-dependent glutaredoxins reduce these disulfide bonds in the cytoplasm and restore protein function [6].

A temperature upshift is one of the first and most reliable cues for pathogens signaling the entry into a warm-blooded host from a comparatively cool environment. Accordingly, many pathogenic bacteria induce the expression of virulence genes at 37°C [7–9]. This also applies to *Y. pseudotuberculosis* and its close relative *Yersinia pestis*. LcrF, the master regulator of virulence, is induced at 37°C by multiple means. At ambient temperatures, transcription of *lcrF* is repressed by the nucleoid-associated protein YmoA. Upon a temperature upshift, YmoA is degraded by the ClpP and Lon proteases, relieving transcriptional repression [10]. In addition, translation of the *lcrF* mRNA is blocked by an RNA thermometer (RNAT), which sequesters the ribosome binding site (RBS) at low temperatures and thereby prevents access of the ribosome. At 37°C this RNAT melts and releases the RBS, allowing LcrF synthesis [11]. LcrF acts as a transcriptional activator for several virulence-associated genes located on the virulence plasmid pYV, which code for the type III secretion system (T3SS), effector proteins translocated by the T3SS, or the adhesin YadA [12].

Strikingly, these typical virulence genes are far from being the only ones affected by temperatures changes. RNA sequencing revealed more than 300 differentially transcribed genes

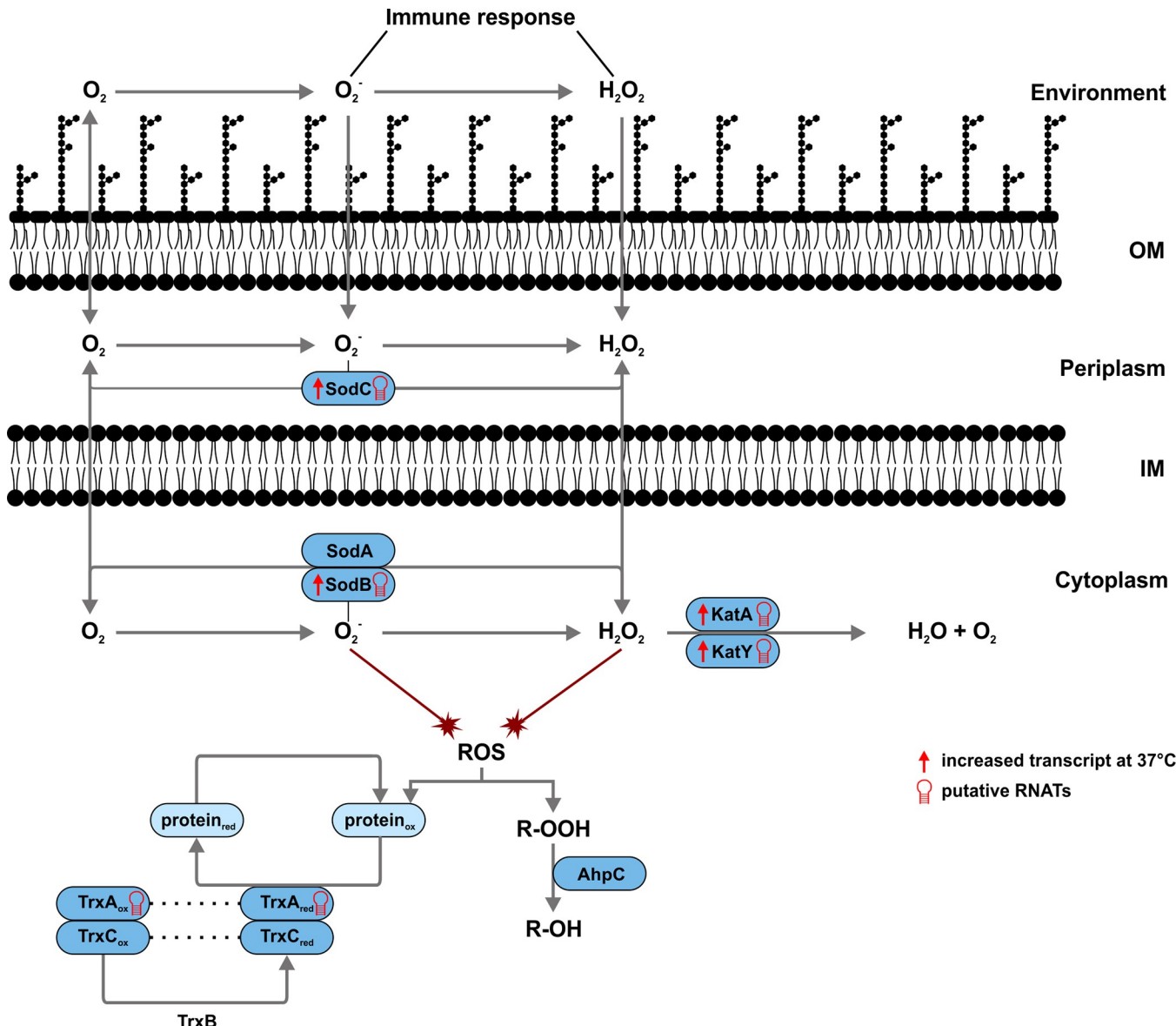

**Fig 1. ROS detoxification pathways in *Y. pseudotuberculosis* and potential temperature regulation mechanisms.** Superoxide and hydrogen peroxide are released by the immune response of the host and enter the bacterial cell. The periplasmic superoxide dismutase SodC encounters superoxide first and, together with its cytoplasmic equals SodA and SodB, transforms it into hydrogen peroxide. This is further neutralised by the catalases KatA and KatY into water and oxygen. Proteins, which have been oxidized by ROS, can be reduced by the thioredoxins TrxA and TrxC. TrxB is responsible for returning the thioredoxins to their reduced state. The alkyl hydroperoxide reductase AhpC is furthermore involved in the detoxification of hydrogen peroxide and its derivatives by converting them to water and alcohols. It is the primary scavenger of endogenously generated hydrogen peroxides. Increased transcription [13] or putative translation [14,15] at 37 compared to 25˚C of ROS detoxification genes is indicated.

between 25 and 37˚C [13]. Among them are genes of the oxidative stress response, namely the genes coding for the superoxide dismutases SodB and SodC and the catalases KatA and KatY (Fig 1). Apart from this transcriptional regulation, global RNA structuromics approaches revealed the existence of RNATs upstream of numerous genes suggesting a contribution to sensing and responding to a warm-blooded host [14,15]. Several of the RNAT-regulated genes play a direct role in virulence, such as the genes for the adhesion protein AilA [14], the

virulence factor OmpA [16], the T3SS components YscJ and YscT [17], the T3SS regulator YopN [18], and the secreted bacterial toxin CnfY [19].

As in many other bacteria, the transcriptional regulator OxyR is the master regulator of $H_2O_2$ defense genes in *Yersinia* species [20,21]. The existence of putative RNATs upstream of several oxidative stress response genes, more specifically *sodB*, *sodC*, *katA*, *katY* and *trxA* [14,15] (Fig 1), however, suggested an additional temperature-responsive regulation of ROS defense genes in *Y. pseudotuberculosis*. In support of a role of catalases in the warm-blooded host, previous reports in *Y. pestis* showed that KatY is a temperature-regulated protein [22,23]. The underlying mechanism of preferential *katY* expression at 37˚C remained elusive, but a predicted secondary structure in the 5'-untranslated region (5'-UTR) was taken as evidence for an RNAT-like mechanism [21].

In this study, we investigated transcriptional and translational control mechanisms of various ROS defense genes in *Y. pseudotuberculosis*. We used a broad range of *in vitro* and *in vivo* approaches to tease apart the different control levels. This work provides detailed insights into the regulation of five ROS detoxification genes. We paid particular attention to the biological roles of the catalases KatA and KatY and identified a novel RNAT upstream of *katY*, which is responsible for conferring improved protection against $H_2O_2$ at 37˚C.

## 2 Material and methods

### 2.1 Bacterial strains and plasmids

Bacterial strains used in this study are listed in S1 Table. Cells were grown in lysogeny broth (LB; 1% NaCl, 1% Tryptone and 0.5% yeast extract) at indicated temperatures. Cultures were supplemented with 150 μg/ml ampicillin or 30 μg/ml chloramphenicol when necessary.

### 2.2 Plasmid construction

All used plasmids and oligonucleotides are listed in S2 and S3 Tables, respectively. Point mutations were generated by site-directed mutagenesis according to the instruction manual of the QuikChange mutagenesis kit (Agilent Technologies).

The RNAT-*bgaB* fusion plasmids (pBO4909 and pBO4436) were constructed by amplifying the 5'-UTR including 30 bp of the coding region with the respective primers sodC_UTR_fw/ sodC_UTR_rev and katY_UTR_fw/katY_UTR_rev, digested with NheI and EcoRI and ligated into pBAD2-*bgaB*-His. To introduce mutations into the RNAT-*bgaB* plasmids their respective rep or derep primers were used with the RNAT-*bgaB* plasmids as a template.

The run-off plasmid for *in vitro* transcription of the RNATs were generated by blunt-end ligation of a PCR-amplified DNA fragment (respective primers RNAT_ro_fw/RNAT_ro_rev), containing the T7 promoter, the RNAT and 60 bp of the coding region, into the EcoRV or NaeI restriction site of pUC18. To introduce mutations into the RNAT-runoff plasmids their respective rep or derep primers were used with the RNAT-runoff plasmids as template.

For deletion of the *katA* and *katY* gene, two fragments flanking the target gene, with an overlap to each other, were constructed by PCR with primers katA/Y 5'flank_fw/rev and katA/ Y 3'flank_fw/rev and recombined by SOE-PCR [24]. This fragment was cloned into the pDM4-suicide plasmid after restriction with SalI and XbaI and transferred to *Y. pseudotuberculosis* by conjugation with *E. coli* S17-1 λ-pir. After selection on LB plates with 10% sucrose, chloramphenicol-sensitive and sucrose-resistant colonies were checked for deletion of the target gene using the katA/Y_EP_fw/rev and katA/Y_IP_fw/rev primer pairs and verified by DNA sequencing (pBO6868; Δ*katA* and pBO7212; Δ*katY*).

For complementation of the deletion strains, NEBuilder HiFi DNA Assembly was utilized. The plasmids pBO6868 and pBO7212 were linearized by PCR amplification using the primer pairs pDM4_katA/Y_Del_fw and pDM4_katA/Y_Del_rev, respectively.

The 5'-UTR with the coding region of *katA* or *katY* and a C-terminal His-Tag were amplified using the primer pairs katA/Y-His_fw and katA/Y-His_rev. In a second amplification, an overlap to the linearized pBO6868 and pBO7212 plasmids was added using the primers katA/Y_pDM4_Del_fw and katA/Y_pDM4_Del_rev. The linearized plasmid and the complementation fragment with the overlap were mixed and assembled, using the NEBuilder HiFi DNA Assembly according to manufacturer's instructions. After successful assembly, the plasmid was transferred into *Y. pseudotuberculosis* by conjugation and after double homologous recombination, colonies were checked for complementation of the target gene using the same primers as for the deletion strains and verified by DNA sequencing (pBO6888; Δ*katA* + *katA*-His; pBO7251; Δ*katY* + *katY*-His). Additionally, the same procedure was used to generate the complementation plasmid pBO7246. Linearization of the pBAD-His A plasmid was achieved by use of the primers pBAD-His_fw/rev and the overlap for the *katY*-His fragment was amplified using primers katY_pBAD_fw and His_pBAD_rev.

## 2.3 Reporter gene activity assay

*E. coli* DH5α or *Y. pseudotuberculosis* YPIII cells carrying the various RNAT-*bgaB* fusion plasmids were inoculated to an optical density at 600 nm ($OD_{600}$) of 0.1. After growth to an $OD_{600}$ of 0.5 at 25°C, transcription was induced with 0.01% in *E. coli* or 0.1% in *Y. pseudotuberculosis*, L-arabinose. The culture was split and shifted to 25 and 37°C. After incubation for 30 min, 400 μl samples were subsequently taken for β-galactosidase assay, 2 ml samples for Western blotting and 4 ml samples for RNA isolation. The β-galactosidase assay was carried out as described previously [25].

## 2.4 Western blot analysis

Cell pellets were resuspended according to their optical density (100 μl per $OD_{600}$ of 1) in 1 x SDS sample buffer (2% SDS, 0.1% bromophenol blue, 1% 2-mercaptoethanol, 25% glycerol, 50 mM Tris/HCl, pH 6.8). The samples were boiled for 10 min at 95°C and after centrifugation (10 min, 13000 rpm), the supernatant was loaded and separated by SDS gel electrophoresis in 5% stacking and 12% separating gels. By tank blotting, the proteins were transferred onto a nitrocellulose membrane (Hybond-C Extra, GE Healthcare) and an anti-His-HRP conjugate antibody (Bio-Rad) was used in a 1:4000 dilution. Chemiluminescence signals were detected by incubating membranes with Immobilon Forte Western HRP substrate (Millipore) with a ChemiDoc MP Imaging System (Biorad).

## 2.5 Quantitative western blot analysis

Samples were prepared as described above. The supernatant was separated by SDS gel electrophoresis in TGX Stain-Free FastCast 12% Acrylamide Gels (Biorad). After separation, the Stain-Free visualization was enabled by activation of the gels for 45 seconds with UV-light before the proteins were transferred by Trans-Blot Turbo Transfer (Biorad) onto a nitrocellulose membrane (Trans-Blot Turbo RTA, Biorad). A mouse anti-His antibody (Biorad) was used as a primary antibody in a 1:500 dilution. A goat anti-mouse IgG StarBright Blue 700 fluorescent antibody (Biorad) was used as a secondary antibody in a 1:2500 dilution. Fluorescence signals and Stain-Free Imaging were detected in a ChemiDoc MP Imaging System (Biorad). The fluorescence signals were quantified by normalization to the total protein

amount detected by Stain-Free visualization, after determining the linear range using Image Lab (Biorad).

## 2.6 RNA extraction and quantitative reverse transcription PCR (qRT-PCR)

Using the peqGOLD Trifast reagent according to the manufacturer's protocol, total RNA was extracted. RNA samples were treated with DNase (TURBO DNA-free Kit, Invitrogen) to remove DNA contamination. Synthesis of cDNA was performed using the iScript cDNA synthesis Kit (Bio-Rad) according to the manufacturer's protocol with 1 μg RNA per reaction. 2 μl of 1:10 diluted cDNA were mixed with 250 nM of each primer, 5 μl of 2x iTaq Universal SYBR Green Supermix, and 2.5 μl sterile water (Carl Roth). In a CFX Connect Real-Time System (Bio-Rad) the amplification and detection of PCR products was measured. To calculate primer efficiency and determine the linear range of amplification, standard curves were employed. Relative transcript amounts were calculated using the primer efficiency corrected method [26]. The non-thermoregulated reference genes *gyrB* and *nuoB* were used for normalization.

### *2.7 In vitro* transcription

RNA for structure probing and primer extension inhibition experiments were synthesized *in vitro* by run-off transcription with T7 RNA polymerase (Thermo Scientific) from EcoRV- or NaeI-linearized pUC18-RNAT + 60 nt plasmids (listed in S2 Table) as previously described [14].

## 2.8 Enzymatic RNA structure probing

RNA structure probing of the 5'-UTR and 60 nt of the RNATs was performed with *in vitro* transcribed RNA. 5'-[$^{32}$P]-labeled RNA (30000 cpm) was mixed with buffer and tRNAs, preincubated for 5 min at various temperatures and treated with T1 (0.0017U) (Invitrogen) or T2 (0.075U) (MoBiTec) RNases for 5 min. For digestion, 5x TN buffer (100 mM Tris acetate, pH 7, 500 mM NaCl) was used. An alkaline hydrolysis ladder and T1 ladder were prepared as described in [27]. All reactions were stopped by addition of formamide loading dye and boiling at 95˚C. Samples were separated on an 8–12% denaturing polyacrylamide gel.

## 2.9 Primer extension inhibition analysis (toeprinting)

Toeprinting analysis was performed with 30S ribosomal subunits, *in vitro* transcribed RNA and tRNA$^{fMet}$ (Sigma-Aldrich) according to [28]. The 5'-[$^{32}$P]-labeled oligonucleotide "gene"_ro_rv, complementary to the 3'end of the *in vitro* transcribed RNA of interest, was used as a primer for cDNA synthesis. The radiolabeled primer (0.16 pmol) was annealed to the RNAT-mRNA (0.08 pmol), incubated with 30S ribosomal subunits (12 pmol) or Tico buffer (60 mM HEPES/KOH, 10.5 mM Mg(CH$_3$COO)$_2$, 690 mM NH$_4$COO, 12 mM 2-mercaptoethanol, 10 mM spermidine, 0.25 mM spermine) in presence of tRNA$^{fMet}$ (8 pmol) at 25˚C, 37˚C or 42˚C for 10 min. Synthesis of cDNA was performed for 10 min at 37˚C after addition of 2 μl MMLV-Mix (VD+Mg$^{2+}$ buffer, BSA, dNTPs and 800 U MMLV reverse transcriptase (Invitrogen). The reaction was stopped by addition of formamide loading dye and boiling at 95˚C. Samples were separated on an 8–12% denaturing polyacrylamide gel. The Thermo Sequenase cycle sequencing Kit (Applied Biosystems) was used for sequencing reactions with the pUC18-RNAT+ 60 bp plasmids as template and radiolabeled primer "gene"_ro_rv.

## 2.10 Zone of inhibition assay

Overnight cultures of various *Y. pseudotuberculosis* strains, grown at 25 and 37˚C were diluted to an optical density at 600 nm (OD$_{600}$) of 0.1. A total of 100 μl was added to 5 ml of soft agar,

briefly mixed and poured on 15 ml LB plates. A Whatman paper disk was applied to the center of the plate and loaded with 3 μl of 5.5 M $H_2O_2$. After incubation for 24 h at 25 and 37°C, the zone of inhibition was measured.

## 2.11 Growth under various $H_2O_2$ concentrations

Cells of the early exponential phase grown at 25 and 37°C were diluted to an optical density at 600 nm ($OD_{600}$) of 0.05 and 100 μl were transferred into a clear 96-well plate. 1 μl of various $H_2O_2$ concentrations were added to the wells to the indicated concentration. Growth was recorded over 20 h at 25 and 37°C.

## 2.12 Measurement of roGFP2-based probe oxidation state

Oxidation of roGFP2-Orp1 was measured as described in [29] and normalized according to [30] for OxD calculation with slight variation in the experimental setup. Briefly, *Y. pseudotuberculosis* cells of various strains harboring the roGFP2 plasmids were inoculated to an optical density at 600 nm ($OD_{600}$) of 0.1 and grown at 25 and 37°C. After growth to an $OD_{600}$ of 0.5, expression was induced with 100 μM IPTG and incubated overnight at 25 or 37°C. Cells were washed twice in PBS and resuspended in PBS to an $OD_{600}$ of 0.2 before 100 μl were transfered into a black, clear-bottom 96-well plate (Nunc, Thermo Scientific). Fluorescence intensities were recorded over 10 min in a microplate reader (Infinite M Plex, Tecan) at the excitation wavelengths 405 and 488 nm and emission wavelength of 530 nm at room temperature. Afterwards, 1 μl of 100 mM AT-2 (2,2′-Dipyridyl disulfide), 1 M DTT (Dithiothreitol) or $H_2O_2$ to the indicated concentrations were added. Changes in fluorescence intensities were measured for 2 h.

## 2.13 Catalase activity assay

For determining catalase activity, decomposition of $H_2O_2$ was measured over time by UV-light according to [31]. Early-exponential cells were harvested after growth at 25 or 37°C, resuspended in phosphate buffer (degassed 17 mM sodium phosphate buffer, pH 8.3) and lysed by ultrasonication. Cell debris were removed by centrifugation for 20 min at 21.000 g and 4°C. After Bradford determination, 50 μg of supernatant proteins were mixed with phosphate buffer to a volume of 180 μl. The absorption at 240 nm was monitored for 1 min at 25°C in a quartz cuvette before adding 20 μl of 0.1 M hydrogen peroxide. As a blank, phosphate buffer mixed with $H_2O_2$ was used. After addition, the absorption of $H_2O_2$ was continuously monitored, until the reaction left the linear range. The velocity of $H_2O_2$ decomposition was calculated based on the linear range.

# 3 Results

## 3.1 Elevated transcription of several oxidative stress response genes at 37°C

A previously conducted transcriptome study revealed that multiple oxidative stress response genes are upregulated at 37°C compared to 25°C [13]. We compared these findings with recent RNA-seq data from our group (S1 Data) and validated the transcription of four genes (*sodB*, *sodC*, *katA* and *katY*) by qRT-PCR. Consistent with the published results, some but not all genes involved in ROS detoxification showed an upregulation at 37°C (Fig 2). Apart from the transcripts encoding the superoxide dismutases SodB and SodC and the catalase KatA, the *katY* mRNA stood out with an about 30-fold induction in all three experimental approaches.

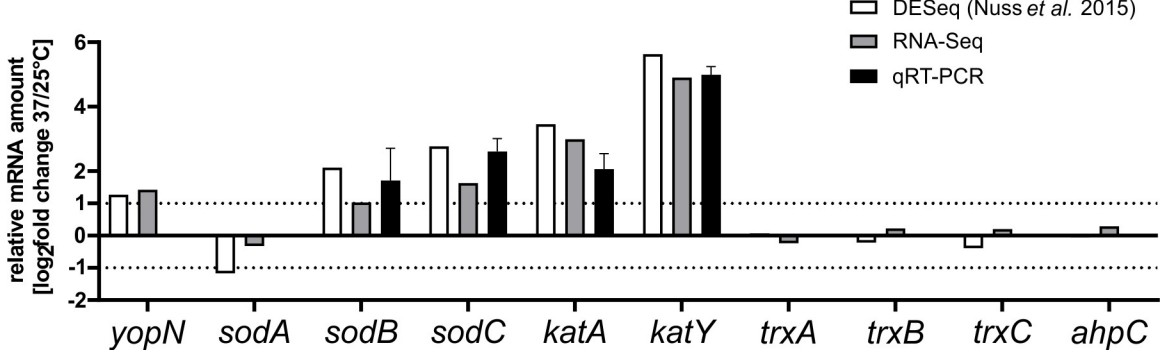

**Fig 2. Transcription of four ROS detoxification genes is elevated at 37˚C.** Relative transcript levels (37/25˚C) are shown. White columns represent differential expression RNA-Seq (DESeq) data from [13]. Grey columns represent RNA-Seq data from this study. Black columns represent qRT-PCR results of selected targets (*sodB*, *sodC*, *katA* and *katY*). RNA was isolated from exponential *Y. pseudotuberculosis* cells, constitutively grown at 25 and 37˚C in biological triplicates. qRT-PCR results were normalized to *gyrB* and *nuoB* as reference genes. The mean and corresponding standard deviation are shown. The *yopN* transcript is shown as a reference for a proven RNAT-regulated gene [18].

### 3.2 The 5'-UTRs of oxidative stress response genes contain putative RNATs

Various transcriptional and post-transcriptional mechanisms can account for the upregulation of ROS defense genes under virulence conditions. Previous global structure probing approaches suggested at least some contribution of translational control by temperature-modulated RNATs [14,15]. To validate and extend these findings, we translationally fused candidate 5'-UTRs to *bgaB* encoding a heat-stable and His-tagged beta-galactosidase and measured its activity and protein amount at 25 and 37˚C (Fig 3A). This reporter gene system uncouples the native transcription from translation since transcription is controlled by the arabinose-inducible pBAD promoter. The recently described *yopN* RNAT [18] served as positive control. The *sodA* gene, which does not contain an obvious RNAT in its 5'-UTR and is barely temperature-regulated at the translational level [14] was chosen as negative control.

First, we investigated the putative RNATs for their ability to control translation in *E. coli* as a host (Fig 3B). For the catalase genes *katA* and *katY*, we observed a pronounced increase in beta-galactosidase activity and protein amount at 37˚C. This was much less evident for the *sod* genes, in particular *sodA* (negative control) and *sodB*, where the enzyme activities increased about 2.5-fold. Since essentially all cellular processes in *E. coli* are more efficient at 37˚C than at 25˚C, we typically do not consider induction factors below three as significant. The *trxA* gene is an exceptional case. The gene is transcribed from two alternative start sites, leading to a short (58 nucleotides) and a much less abundant long (98 nucleotides) 5'-UTR (S1 Fig). As observed previously [14], only the 5'-UTR of the short transcript contains a functional RNAT as shown by the beta-galactosidase activity and protein increase in case of the short but not the long construct (Fig 3B).

To determine the influence of these RNATs in the native background, we introduced the translational fusions into *Y. pseudotuberculosis*. Unexpectedly, *sodB* behaved like the negative control *sodA*, and the protein levels of the superoxide dismutases were lower at the higher temperature (Fig 3C). Fully consistent with the results in *E. coli*, the 5'-UTR of the long *trxA* transcript showed no RNAT activity, whereas the 5'-UTRs of *katY* and the short *trxA* transcript conferred translational repression at 25˚C and induction at 37˚C, most prominently again for *katY* with a more than 20-fold change.

To ascertain that the mRNA levels derived from the pBAD promoter were roughly similar at 25 and 37˚C, *bgaB* levels were determined by qRT-PCR using the same *Yersinia* samples as

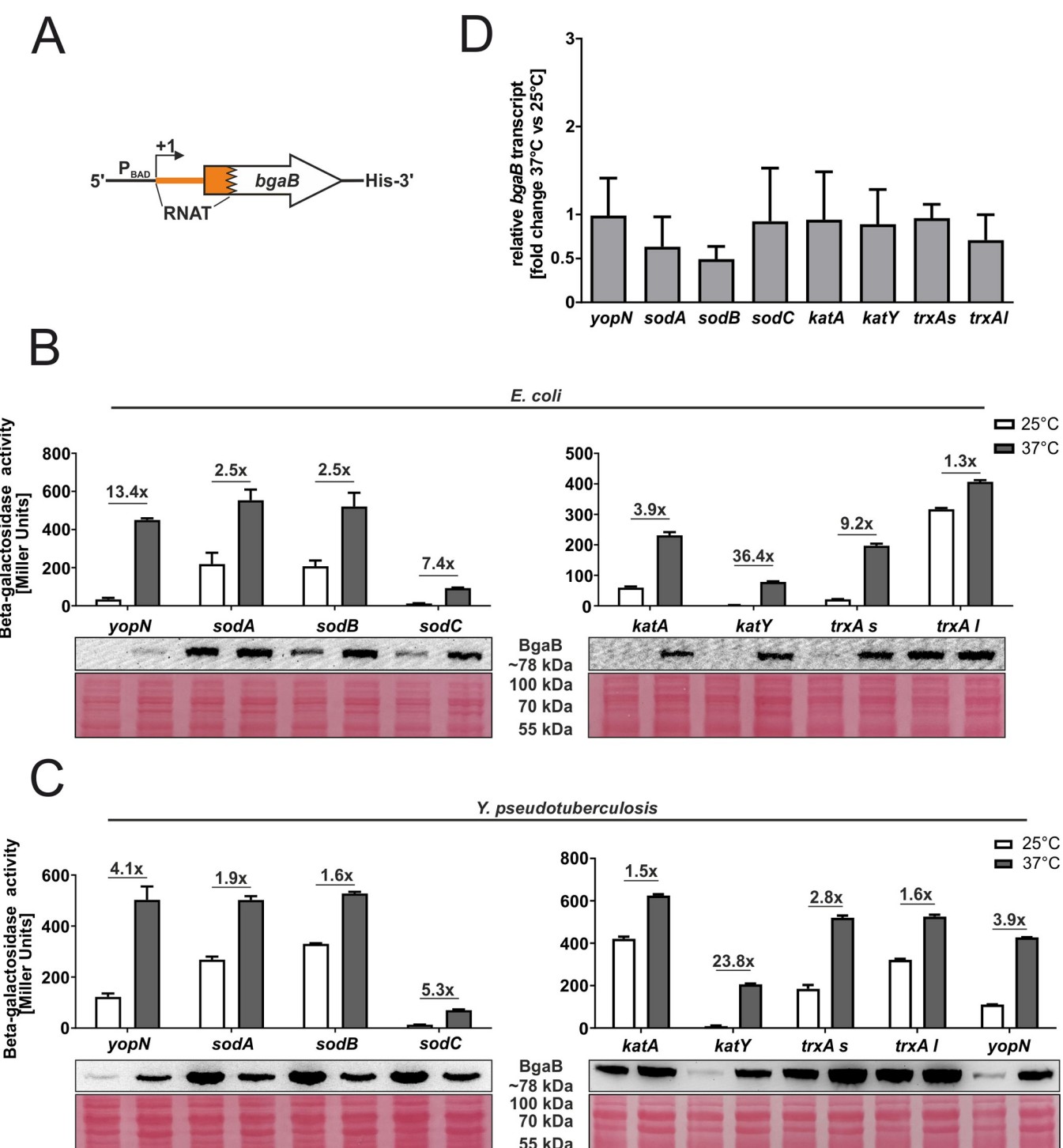

**Fig 3. Translation of *katY* and the short *trxA* isoform is upregulated at 37˚C in *Y. pseudotuberculosis*.** (**A**) Schematic representation of the reporter gene fusion. The RNAT was translationally fused to the *bgaB* gene. Transcription was dependent on the pBAD promoter. As a control, the *yopN* RNAT was used. The fusion plasmids were introduced into *E. coli* (**B**) or *Y. pseudotuberculosis* YPIII cells (**C**) and grown to an $OD_{600}$ of 0.5 at 25˚C. Transcription from the pBAD promoter was induced by the addition of 0.01% or 0.1% L-arabinose in *E. coli* and *Y. pseudotuberculosis*, respectively. The cultures were split and incubated at 25 or 37˚C. After 30 min, samples were taken for β-galactosidase assays, Western blot analysis and qRT-PCR. Experiments were carried out multiple times. Mean and corresponding standard deviation of biological triplicates are shown. Western blot membranes were stained with Ponceau S as a loading control. One representative Western blot is shown. (**D**) Levels of *bgaB* transcript determined by qRT-PCR from cells used in (**C**) were normalized to *gyrB* and *nuoB* mRNA amounts. The mean of three biological replicates and technical triplicates with their corresponding standard deviation are shown.

in Fig 3C. Comparison between 25 and 37°C showed almost equal mRNA levels for most constructs (Fig 3D). Only the *sodA* and *sodB* fusions showed a reduced *bgaB* transcript amount at 37°C, which might at least in part explain the reduced BgaB-His protein levels (Fig 3C). Overall, these results support the hypothesis that the 5'-UTRs of several *Yersinia* ROS defense genes can regulate translation initiation.

### 3.3 A stabilizing point mutation prevents RNAT regulation of the short *trxA* transcript

The *trxA* transcript caught our attention as it occurs in two isoforms of different lengths (Fig 4A and 4B), of which only the short 5'-UTR acts as translational control element (Fig 3). It folds into a structure that partially occludes the ribosome binding site (Fig 4A). The unpaired adenosine residues in the SD sequence and the start codon might be responsible for

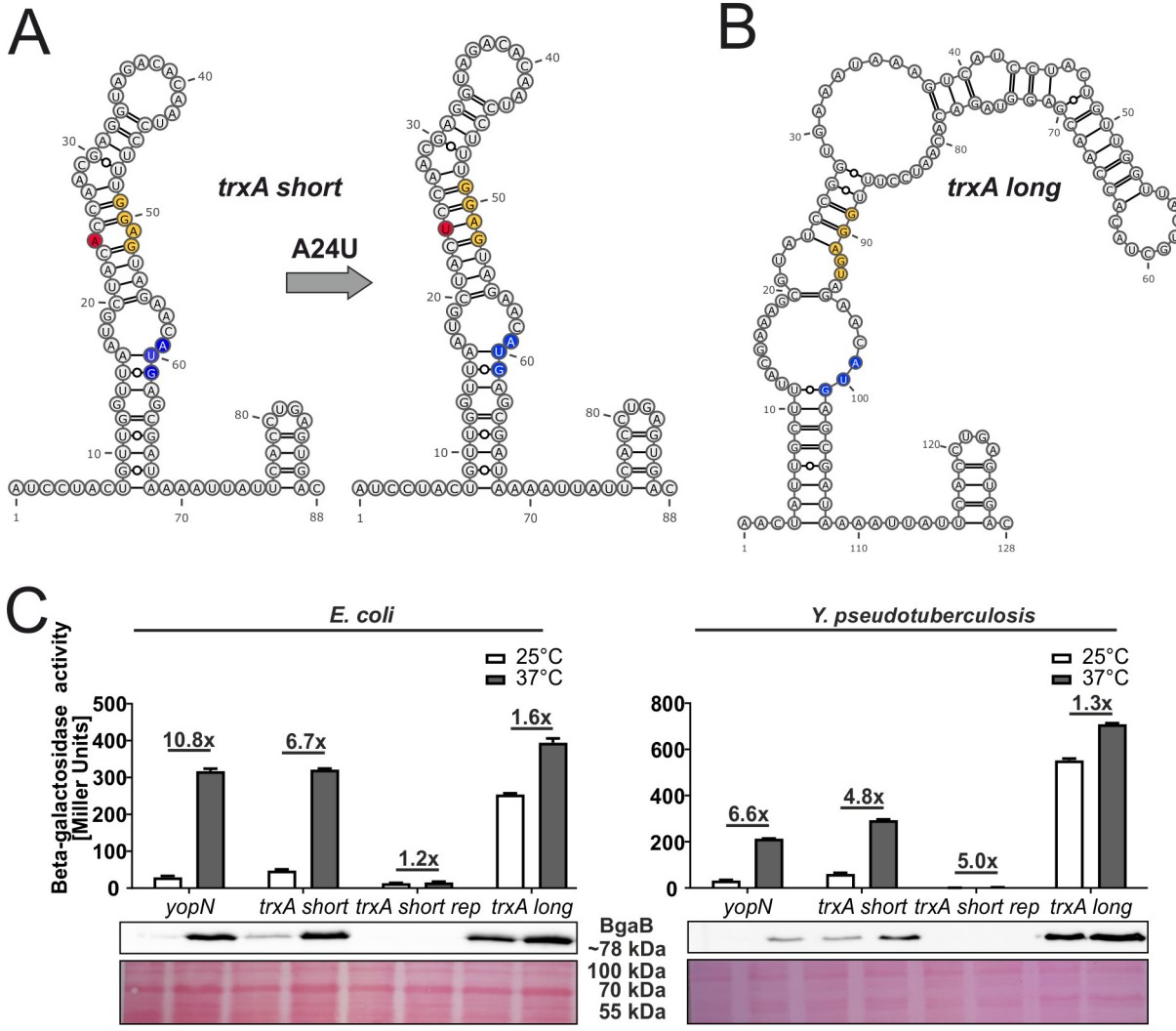

**Fig 4. Only the 5'-UTR of the short *trxA* transcript contains a thermoresponsive RNA structure.** PARS-derived secondary structures of the short (**A**) and long (**B**) *trxA* RNAT at 37°C and the predicted stabilized structure after mutation of the anti-SD sequence. The SD sequence is highlighted in yellow, its corresponding start codon in blue and the mutation site in red.–, AU pair; =, GC pair; and ○, GU pair. (**C**) Translational control was measured by *bgaB* fusions under control of the pBAD promoter. The *yopN* RNAT served as positive control. Experiments were carried out as described in Fig 3.

the temperature responsiveness. Several reasons might account for the high expression of the long *trxA* variant already at low temperature. In contrast to the single mismatch in the SD sequence of the short isoform, the long one features two mismatched residues resulting in a kinked structure (Fig 4B). In addition, the start codon in the long transcript is more accessible than in the short one. Altogether, these differences likely explain why the long version is unable to repress ribosome binding and translation initiation at low temperature. Please note that the long transcript most likely plays a minor role in *Y. pseudotuberculosis* as it is much less abundant than the short transcript at both temperatures (S1 Fig).

To investigate the temperature-responsive RNA structure of the short *trxA* isoform in more detail, we constructed a stabilized version (rep) by replacing the unpaired adenine in the anti-SD sequence (the 5'-UTR sequence that is predicted to pair with the SD sequence) by an uracil resulting in a perfectly paired SD sequence (Fig 4A). In the *bgaB* reporter system, the stabilizing mutation abolished expression both at low and at high temperatures in *E. coli* and in *Y. pseudotuberculosis* (Fig 4C).

To solidify the claim that the short *trxA* RNA structure melts between the SD sequence and the start codon, we applied enzymatic RNA structure probing and treated *in vitro* transcribed 5'-end labeled RNA at different temperatures with RNases T1 (cuts unpaired guanines) and T2 (preferentially cuts unpaired adenines but also other unpaired nucleotides). In the wild-type (WT) RNA, we observed almost no cleavage around the SD sequence and the start codon at 25°C but prominent cleavage at 37 and 42°C consistent with thermally induced melting of these regions (Fig 5A). In accordance with the absence of any reporter gene activity (Fig 4C), the same regions in the rep variant were completely protected from ribonucleolytic attack indicating that melting of the structure is prevented by locking the structure in a closed conformation (Fig 5A).

Finally, we tested the RNAs for their accessibility to ribosome binding by employing primer extension inhibition (toeprinting) assays at different temperatures. The WT and rep RNAs were reversely transcribed in the presence or absence of 30S ribosomal subunits. The occurrence of prematurely terminated reverse transcripts (toeprints) indicates successful binding of the 30S ribosome by acting as a roadblock for cDNA synthesis. As expected, the WT *trxA* transcript generated a toeprint signal in the presence of 30S subunits with increasing temperatures whereas the rep version did not (Fig 5B). Cumulatively, the *in vivo* and *in vitro* experiments provide compelling evidence that the short *trxA* isoform contains a functional RNAT that is solely responsible for temperature regulation since transcription of *trxA* is similar at 25 and 37°C in *Y. pseudotuberculosis* (Fig 2).

### 3.4 Stabilizing mutations in the 5'-UTRs of *sodB*, *sodC* and *katA* impair translation

Despite initial evidence for RNATs in the *sodB* and *sodC* (but not *sodA*) transcripts by global RNA structure profiling through parallel analysis of RNA structures (PARS) and reporter gene assays in *E. coli* [14], the results presented in Fig 3 raised doubts whether they function as thermosensors in *Y. pseudotuberculosis*. We wanted to understand this discrepancy and constructed repressed versions of the *sodB* and *sodC* 5'-UTRs by introducing stabilizing point mutations in the anti-SD region (Fig 6A and 6B). As an indication that the WT structure might have a functional role, the mutated versions prevented translation at both temperatures in *E. coli* and *Y. pseudotuberculosis* (Fig 6C).

Consistent with this assumption, *in vitro* transcribed RNA showed a temperature-induced melting around the RBS whereas the same region in the mutated RNA was protected from ribonucleases even at 42°C except for the bulged nucleotide G85 (Fig 7A). Accordingly,

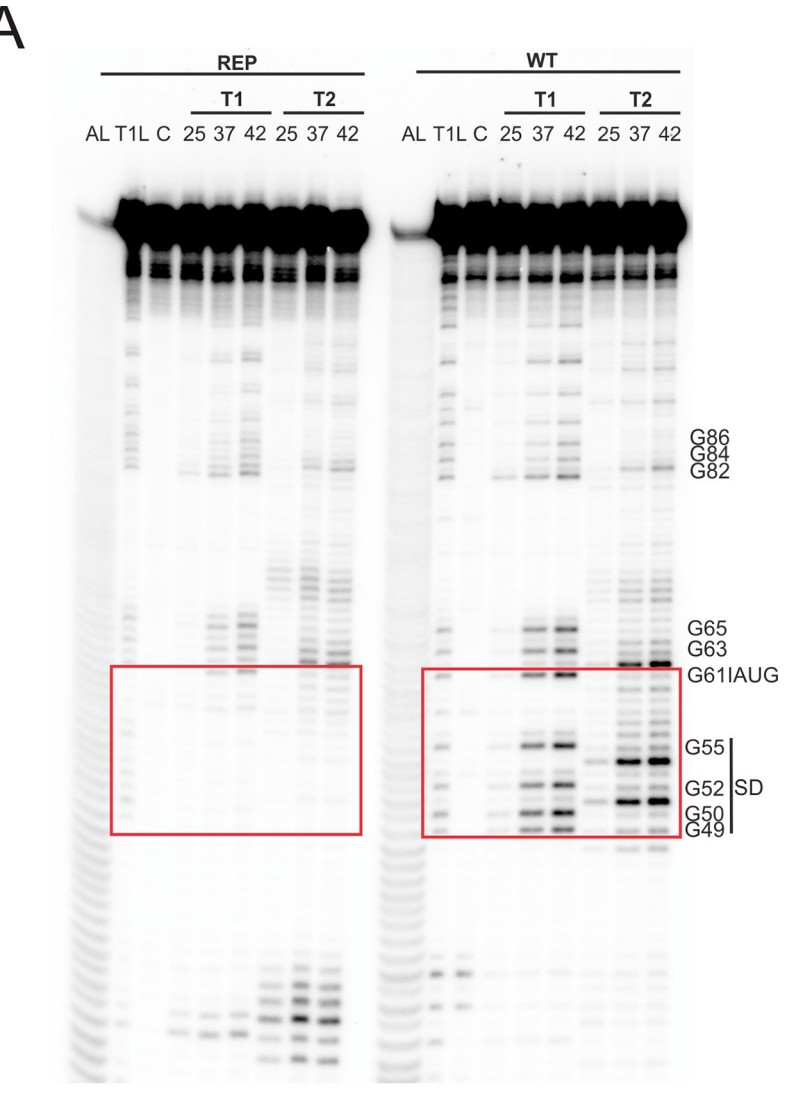

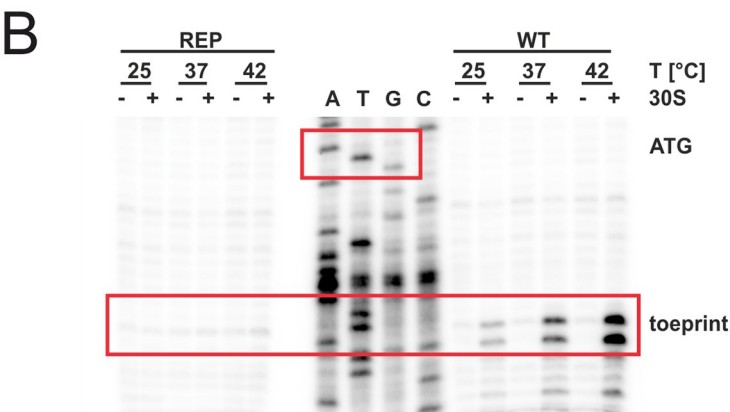

**Fig 5. The short *trxA* RNAT melts at higher temperatures and facilitates ribosome binding. (A)** Enzymatic structure probing of the short *trxA* RNAT (WT) and its stabilized version (REP). Radiolabelled RNA was treated with RNases T1 and T2 at 25, 37 and 42°C. AL, alkaline ladder; T1L, RNase T1 cleavage ladder in sequencing buffer at 37°C; C, RNA treated with water instead of RNases–cleavage control. The ribosome-binding site is highlighted by a red box. **(B)** Primer extension inhibition of the short *trxA* RNAT (WT) and its stabilized version (REP) was conducted at 25,

37˚ and 42˚C with (+) and without (−) the addition of 30S ribosomal subunits. Ribosome binding leads to the accumulation of a toeprint signal. ATGC lanes indicate sequencing reactions for orientation. Position of ATG and the toeprint signal is highlighted by red boxes. Experiments were carried out at least twice.

toeprinting revealed ribosome binding to the WT *sodB* RNAT at 37 but not at 25˚C, and no toeprint signal was observed for the stabilized version (Fig 7B). Similar observations were made when the *sodC* 5'-UTRs (WT and rep variant) were subjected to structure probing (Fig 7C) and toeprinting (Fig 7D). Altogether, these biochemical experiments support the idea that the *sodB* and *sodC* 5'-UTRs are able to melt in response to increasing temperature to permit ribosome binding. In *Yersinia*, however, this response seems to be counteracted by yet unknown mechanisms.

Likewise, we wanted to understand the observed difference in the behaviour of the *katA* 5'-UTR in *E. coli* and *Y. pseudotuberculosis* (Fig 3). Due to the nature of the RNA structure with

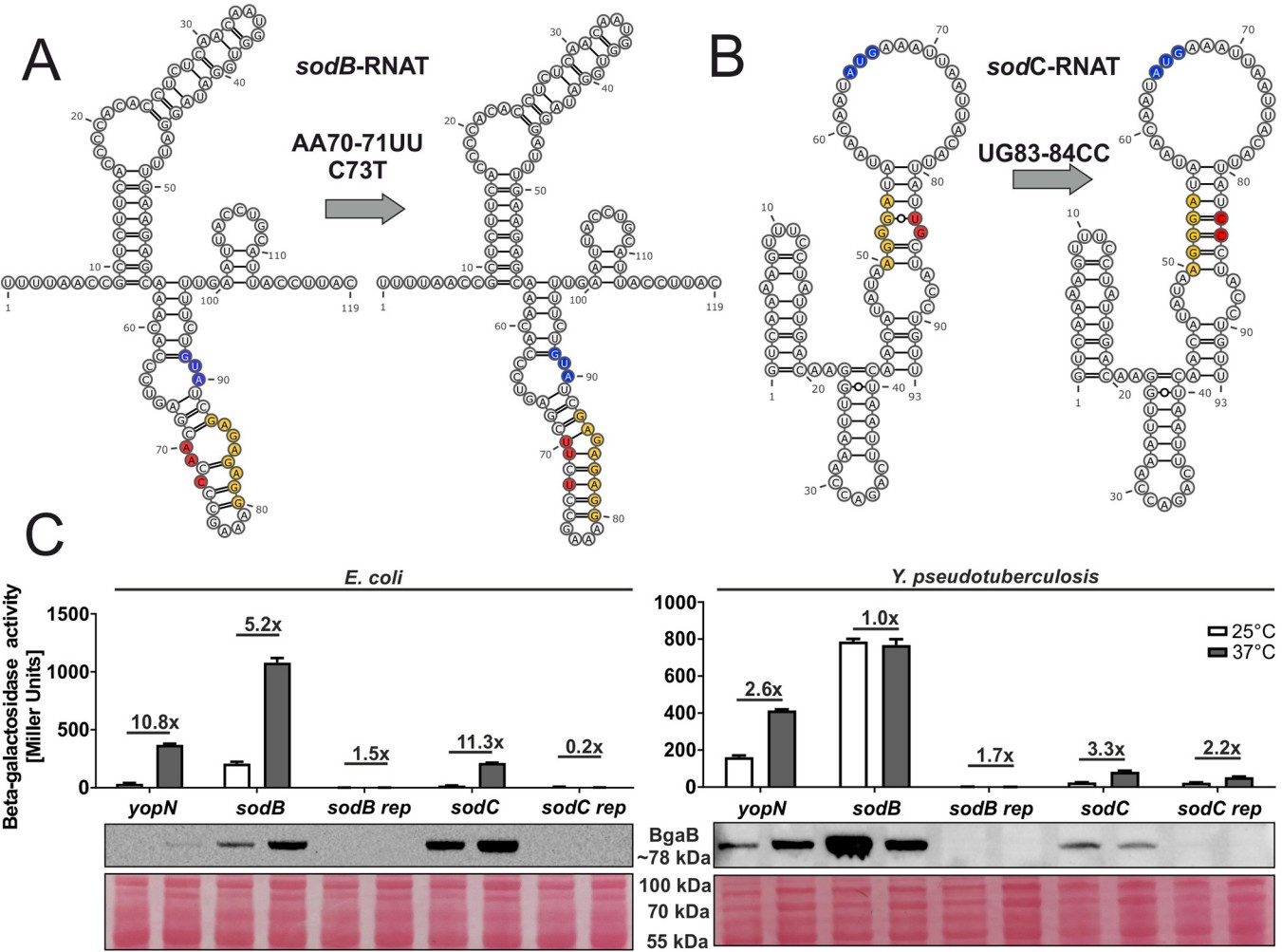

**Fig 6. The 5'-UTRs of the *sodB* and *sodC* transcripts contain a thermoresponsive structure.** PARS-derived secondary structure of the *sodB* RNAT (**A**) and the MFE structure of the *sodC* RNAT (**B**) at 25˚C with their corresponding predicted stabilized structure after mutation of the anti-SD sequence. The SD sequence is highlighted in yellow, the AUG codon in blue and the mutation site in red.–, AU pair; =, GC pair; and ○, GU pair. (**C**) Translational control was measured by *bgaB* fusions. The RNAT was translationally fused to *bgaB* under control of the pBAD promoter. The *yopN* RNAT was used as positive control. Experiments were carried out as described in Fig 3.

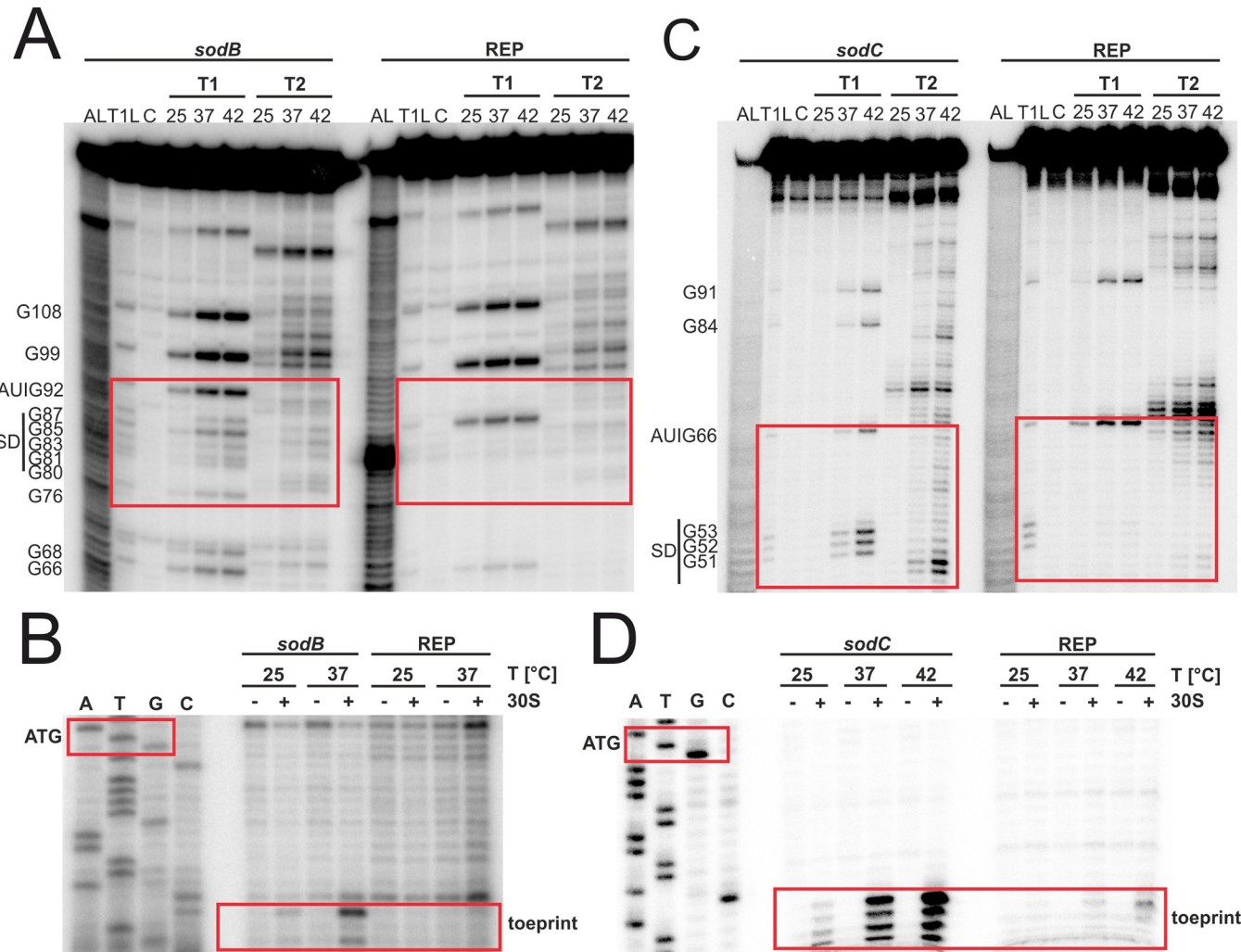

**Fig 7. Structure probing and toeprinting support temperature-responsive melting and ribosome binding to the *sodB* and *sodC* 5'-UTRs.** Enzymatic structure probing of the *sodB* (**A**) and *sodC* (**C**) RNATs and their stabilized version (REP). Radiolabelled RNA was treated with RNases T1 and T2 at 25, 37 and 42°C. AL, alkaline ladder; T1L, RNase T1 cleavage ladder in sequencing buffer at 37°C; C, RNA treated with water instead of RNases–cleavage control. The ribosome-binding site is highlighted by a red box. Primer extension inhibition of the *sodB* (**B**) and *sodC* (**D**) RNATs and their stabilized version (REP) was conducted at 25, 37° and 42°C with (+) and without (−) the addition of 30S ribosomal subunits. Ribosome binding leads to the accumulation of a toeprint signal. ATGC lanes indicate sequencing reactions for orientation. Position of ATG and the toeprint signal is highlighted by red boxes. Experiments were carried out at least twice.

the SD sequence in a terminal loop (Fig 8A), we did not change the anti-SD sequence but stabilized the adjacent hairpin structure by changing the unpaired adenosine and cytosine at position 109 and 110 into a cytosine and guanine, respectively. In the reporter gene assay, we observed the expected effect of translational repression in *E. coli* and *Y. pseudotuberculosis* (Fig 8B). By RNA structure probing, we could further show that the 5'-UTR of *katA* melts at higher temperatures, not only between the SD sequence and the start codon, but also further downstream. According to the PARS profiles at different temperatures, the nucleotides 130–136 should primarily be in a single-stranded conformation prone to RNase cleavage at 25°C [14]. As these residues were partially protected from T1 and T2 attack at low temperature, they are probably engaged in a more complex overall structure. As expected, the rep variant showed almost no cleavage around the entire RBS suggesting substantial stabilization due to the

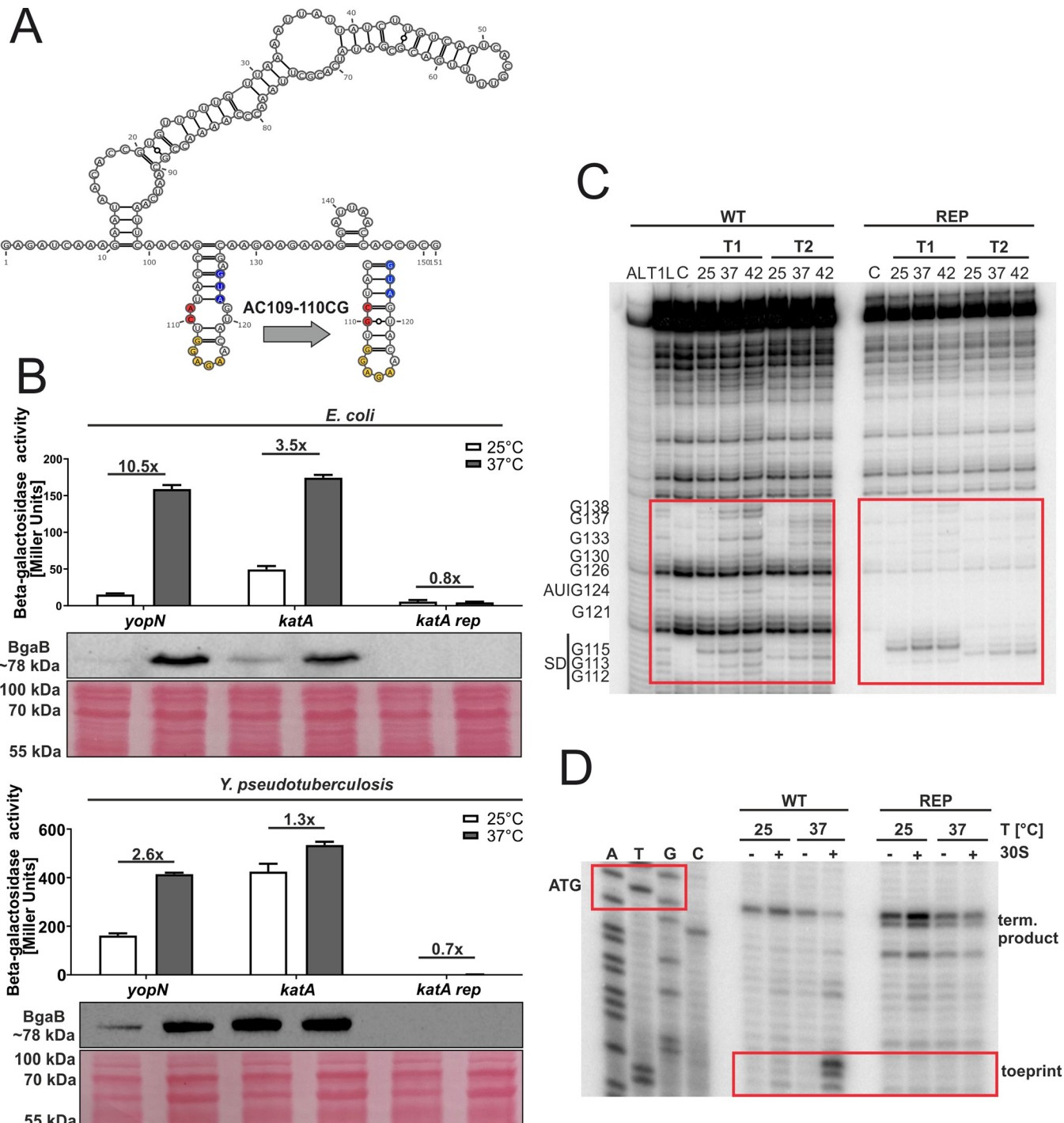

**Fig 8. The 5'-UTR of the *katA* transcript contains a thermoresponsive structure.** (A) PARS-derived secondary structure of the *katA* RNAT at 37°C with its predicted stabilized structure after mutation of a loop in the RBS to create a hairpin structure. The potential SD sequence is highlighted in yellow, its corresponding AUG in blue and the mutation site in red.–, AU pair; =, GC pair; and ○, GU pair. (B) Translational control was measured by *bgaB* fusions. The RNAT was translationally fused to *bgaB* under control of the pBAD promoter. As a control the *yopN* RNAT was used. Experiments were carried out as described in Fig 3. (C) Enzymatic structure probing of the *katA* RNAT (WT) and its stabilized version (REP). Radiolabelled RNA was treated with RNases T1 and T2 at 25, 37 and 42°C. AL, alkaline ladder; T1L, RNase T1 cleavage ladder in sequencing buffer at 37°C; C, RNA treated with water instead of RNases– cleavage control. The ribosome-binding site is highlighted by a red box. (D) Primer extension inhibition of the *katA* RNAT (WT) and its stabilized version (REP) was conducted at 25, 37°C, with (+) and without (−) the addition of 30S ribosomal subunits. Ribosome binding leads to the accumulation of a toeprint signal. ATGC lanes indicate sequencing reactions for orientation. Position of ATG and the toeprint signal is highlighted by red boxes. Experiments were carried out at least twice.

formation of an improved hairpin structure (Fig 8C). Consistent with the probing results, a toeprint signal was found at 37˚C with the WT *katA* RNA, but not with the REP version (Fig 8D). Instead, other premature termination products of reverse transcription were found, which often occur near stable structures in stabilized RNATs [11]. Like *sodB* and *sodC*, the *katA* 5'-UTR appears to be capable of thermally controlling access to the SD sequence, but the dynamic nature of these RNA structures seems to play a minimal role in the *Yersinia* cell under the tested conditions.

## 3.5 The *katY* 5'-UTR harbors an RNAT that is functional in *Y. pseudotuberculosis*

The most prominent temperature response of all ROS defense mRNAs was observed in the case of *katY* (Fig 3). The 5'-UTR of *katY* had escaped our attention as an RNAT candidate in previous RNA structuromic approaches because most of the sequence relevant for masking the RBS is not located in the 5'-UTR but within the coding sequence (Fig 9A). To characterize this new RNAT candidate, we generated structural mutants. A stabilized version (U31C) was constructed by exchanging a weak UG pair in the SD/anti-SD region by a stronger CG pair. This variant completely repressed translation at 37˚C (Fig 9B). Introduction of an additional mutation (UA31-32CU; Fig 9A) initially aimed at stabilizing the structure even further, was predicted to result in a structural re-arrangement that slightly derepressed RNAT activity (Fig 9B). To exclude that these regulatory effects were due to changes in the mRNA levels, we compared the *bgaB* levels from the *Y. pseudotuberculosis* samples used in Fig 9B by qRT-PCR and detected no significant differences between 25 and 37˚C (Fig 9C). In further support of a translational control mechanism, we observed increased sensitivity to RNase cleavage around the SD sequence and the start codon at higher temperatures in structure probing experiments (Fig 9D) and increased binding of the ribosome at higher temperatures in toeprinting assays (Fig 9E). All these results demonstrate that the *katY* 5'-UTR harbors a *bona fide* RNAT.

## 3.6 Temperature affects the susceptibility of *Y. pseudotuberculosis* against $H_2O_2$

The upregulation of *katA* expression at 37˚C on the transcriptional level (Fig 2) and–more prominently–*katY* expression on both the transcriptional and translational level (Figs 2, 3C and 9B), lead us to believe that *Yersinia* cells might be better protected against ROS when grown at 37˚C instead of 25˚C. To test this hypothesis and to dissect the contribution of individual players to ROS detoxification, we generated markerless catalase deletion mutants and tested their susceptibility to $H_2O_2$.

First, we conducted a zone of inhibition assay with *Yersinia* cells pre-grown at 25 or 37˚C and spread onto agar plates. The bacteria were then subjected to a diffusion gradient of $H_2O_2$ from a paper disk, and the plates were incubated at 25 or 37˚C. Congruent with our hypothesis of a better protection at elevated temperatures, we observed a significantly reduced zone of inhibition for the wildtype (WT) strain, if it was grown at 37 compared to 25˚C (Fig 10A). The zone of inhibition at both temperatures became larger when the *katA* gene was removed (Δ*katA*), indicating a general protective effect against $H_2O_2$ as expected for a catalase gene. Deletion of the *katY* gene (Δ*katY*) did not increase the zone of inhibition at 25˚C compared to the WT. Strikingly, the zone of inhibition remained unchanged at 37˚C suggesting that the temperature-induced protection in the WT and Δ*katA* strains is mediated by KatY. Deletion of both catalase genes (ΔΔ*katAY*) resulted in a combined phenotype of the single mutants with an increase in the zone of inhibition and no reduction at 37˚C.

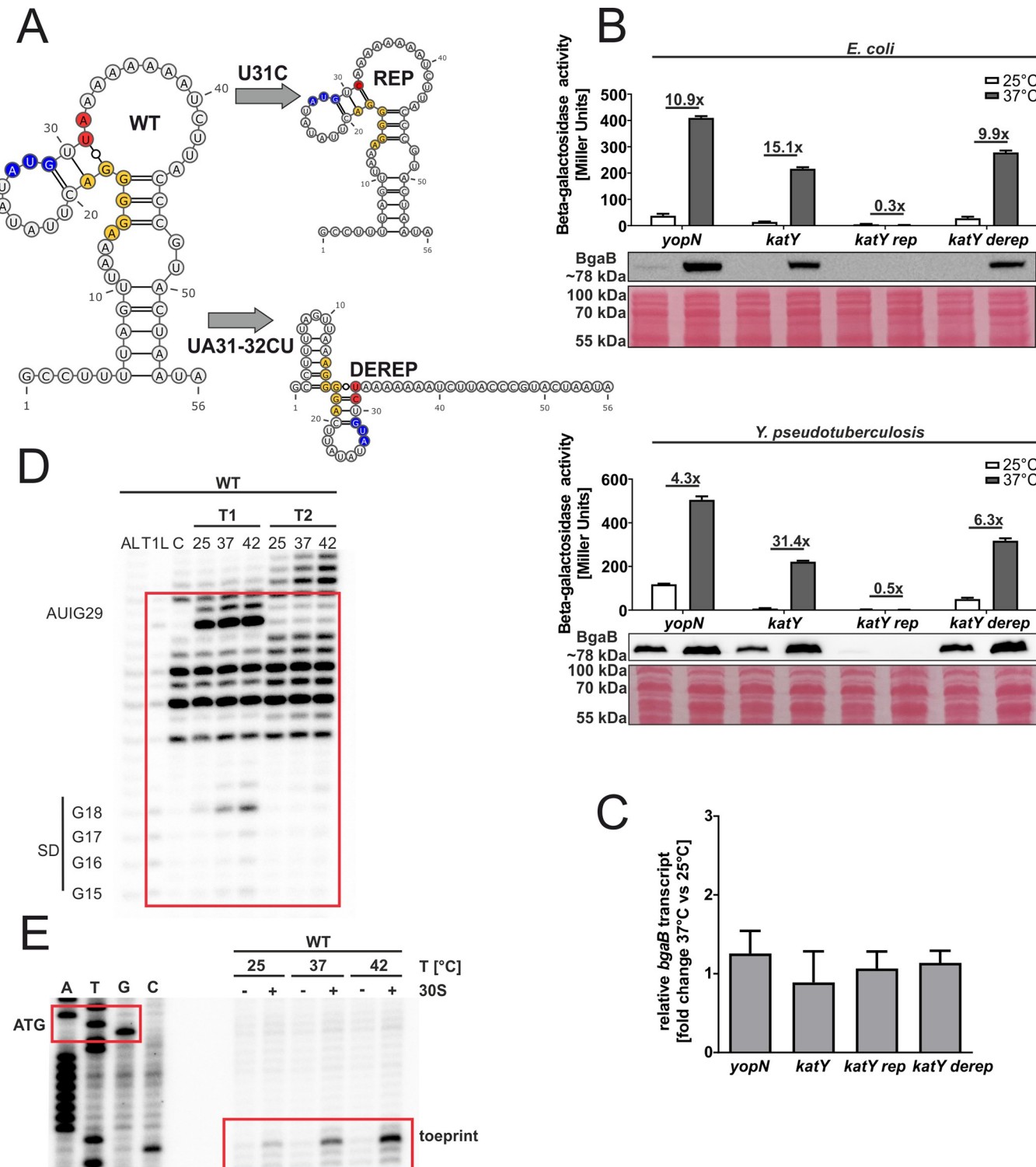

**Fig 9. The *katY* 5'-UTR is a functional RNAT *in vitro* and in *Y. pseudotuberculosis*.** (**A**) MFE-derived secondary structure of the *katY* RNAT at 25˚C with its predicted stabilized (REP) and destabilized structure (DEREP) after mutation of the anti-SD sequence. The potential SD sequence is highlighted in yellow, its corresponding AUG in blue and the mutation site in red.−, AU pair; =, GC pair; and ○, GU pair. (**B**) Translational control was measured by *bgaB* fusions. The RNAT was translationally fused to *bgaB* under control of the pBAD promoter. As a control the *yopN* RNAT was used. Experiments were carried out as described in Fig 3. (**C**) Levels of *bgaB* transcript determined by qRT-PCR from *Y. pseudotuberculosis* cells used in (**B**) were normalized to *gyrB* and *nuoB* mRNA amounts. The mean of three biological replicates and technical triplicates with their corresponding standard deviation are shown. (**D**) Enzymatic

structure probing of the *katY* RNAT (WT). Radiolabelled RNA was treated with RNases T1 and T2 at 25, 37 and 42˚C. AL, alkaline ladder; T1L, RNase T1 cleavage ladder in sequencing buffer at 37˚C; C, RNA treated with water instead of RNases–cleavage control. The ribosome-binding site is highlighted by a red box. **(E)** Primer extension inhibition of the *katY* RNAT (WT) was conducted at 25, 37˚C, with (+) and without (−) the addition of 30S ribosomal subunits. Ribosome binding leads to the accumulation of a toeprint signal. ATGC lanes indicate sequencing reactions for orientation. Position of ATG and the toeprint signal is highlighted by red boxes. Experiments were carried out at least twice.

Next, we tested if these observations hold true in liquid culture by conducting growth experiments under various $H_2O_2$ concentrations and observed similar results, showing reduced susceptibility against $H_2O_2$ at 37˚C and a prominent KatY effect (Figs 10B and S2). Finally, we examined whether protection is due to catalase activity and performed an activity assay testing the ability of cell lysates generated from cells grown at 25 or 37˚C to decompose $H_2O_2$. This assay measures the absorbance of $H_2O_2$ directly by UV-light [31]. Hence, catalase activity correlates with the reduction of absorbance. Consistent with the phenotypes described above, we observed a higher catalase activity if *Yersinia* WT cells were grown at 37˚C compared to 25˚C (Fig 10C). The Δ*katA* mutant, which can only produce KatY, shows essentially no activity at 25˚C but weak activity at 37˚C. The catalase activity of the Δ*katY* mutant at 25˚C was similar to the WT activity suggesting KatA to be the only catalase present at 25˚C. The increase at 37˚C was slightly lower than in the WT. Adding up the activities of both catalases measured at 37˚C in the single mutants resulted in a value similar to the WT activity. As expected, the double mutant ΔΔ*katAY* did not exhibit any catalase activity at all.

Complementing the deletion strains by reintroducing the catalase genes into their native genomic context by homologous recombination (Fig 11A) restored the corresponding phenotypes (S3A and S4 Figs) and catalase activities (S3B Fig). The reintegrated catalase genes encoded a C-terminal His-tag, which allowed us to quantify the KatA and KatY protein amounts at 25 and 37˚C. KatA was present at almost equal amounts at both temperatures in all complemented backgrounds (Fig 11B). KatY, however, was barely detectable at 25˚C and strongly increased at 37˚C, which supports the hypothesis that KatY is the primary temperature-regulated catalase in *Y. pseudotuberculosis*. At least one shorter KatY product was detected in all immunoblots. It presumably represents a truncated KatY derivative as it has been described in *Y. pestis* [22].

In an independent line of complementation experiments, we produced the KatY-His protein from a plasmid under control of the arabinose inducible pBAD promoter. Here also, KatY levels were upregulated at 37˚C, no matter how strongly transcription was enforced by different arabinose concentrations (S5 Fig). In perfect agreement with the results from the *bgaB* fusions, this finding supports the assumption that *katY* expression is to a large extent translationally controlled by an RNAT.

## 3.7 Temperature regulated *katY* expression affects $H_2O_2$ detoxification and intracellular redox state

To further elucidate the interplay between KatA and KatY in temperature-regulated ROS protection, we employed the genetically encoded redox-sensitive probe roGFP2-Orp1, an $H_2O_2$-sensitive variant of roGFP2. The roGFP2 protein carries redox-sensitive cysteines and is a reliable sensor for the intracellular redox-state [32]. roGFP2 has many advantages compared to redox-sensitive fluorescent dyes. For example reversibility, the ability to be genetically targeted into specific cell compartments, and most importantly its quantitative nature due to a ratiometric approach, which compensates for differences of roGFP2 concentration [29]. Here, we used roGFP2-Orp1, which is fused to the oxidant receptor peroxidase Orp1 from yeast, enabling specific and sensitive real-time measurement of intracellular $H_2O_2$ through the

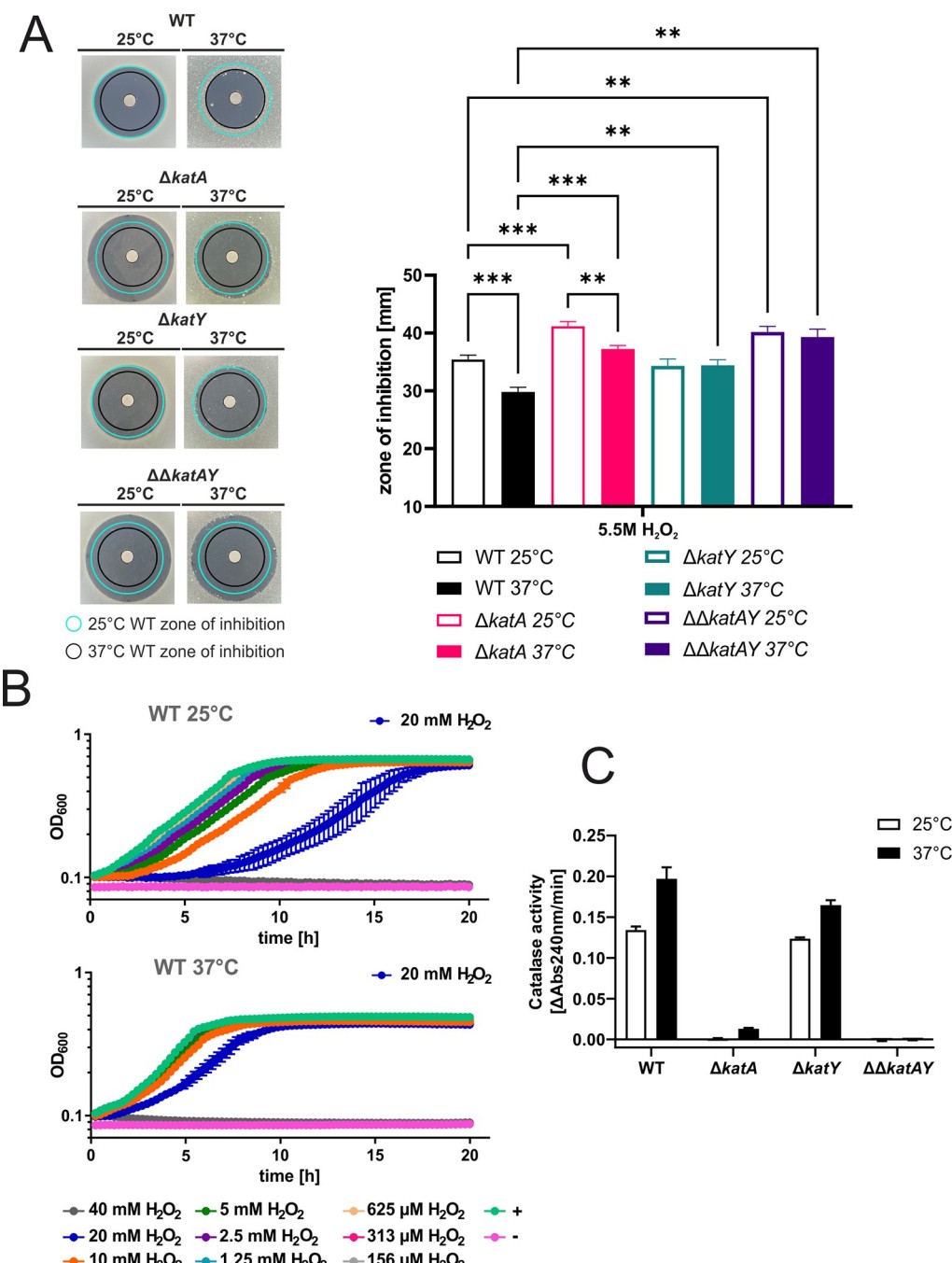

**Fig 10. Temperature-dependent influence of *katA* and *katY* on the oxidative stress response in *Y. pseudotuberculosis*.** (**A**) Disk diffusion assay for $H_2O_2$ was conducted by applying 3 µl of 5.5 M $H_2O_2$ onto paper disks on soft agar containing a bacterial suspension. After 24 h of growth at the indicated temperature the zone of inhibition was measured. The experiment was carried out multiple times and each time in biological triplicates and two technical replicates. Cyan ring = zone of inhibition measured for the wildtype at 25˚C. Black ring = zone of inhibition measured for the wildtype at 37˚C. Asterisks indicate statistically significant differences by oneway ANOVA (n = 3; * p < 0.05; ** p < 0.01; *** p < 0.001) (**B**) Cells were pre-grown at 25˚C or 37˚C and diluted to an $OD_{600}$ of 0.05 in a 96 well plate. $H_2O_2$ was added to the indicated final concentrations. Growth was monitored by measuring the $OD_{600}$ during incubation at 25˚C or 37˚C. The highest concentration, which allowed growth is highlighted. + = no $H_2O_2$ control,— = medium control. The mean and the standard deviation of biological triplicates are plotted. Experiments were carried out multiple times. (**C**) Decomposition of $H_2O_2$ was measured in real-time by reading the absorption of $H_2O_2$ at 240 nm over time at 25˚C. Cells were grown at 25˚C or 37˚C until an $OD_{600}$ of 0.5 was reached. Cells were lysed by ultrasonication. Lysate with a protein concentration of 50 µg/ml, determined by Bradford assay, was used and treated

with 0.01 M $H_2O_2$. The velocity of $H_2O_2$ decomposition was calculated based on the linear range at the beginning of the curve. The mean and standard deviation are shown. Experiments were carried out at least twice in biological triplicates.

efficient proximity-based peroxidase redox relay [29,33]. $H_2O_2$ causes oxidation of the Orp1 thiol groups leading to the formation of disulfide bonds (Fig 12A). Oxidized Orp1 promotes oxidation of the thiol groups of roGFP2 due to its close proximity leading to the formation of disulfide bonds at the roGFP2 protein. This formation induces a small conformational change in the protein resulting in a shift in the excitation spectrum. Under reduced conditions (untreated or DTT; dithiothreitol treated), roGFP2 exhibits an excitation peak at around 480

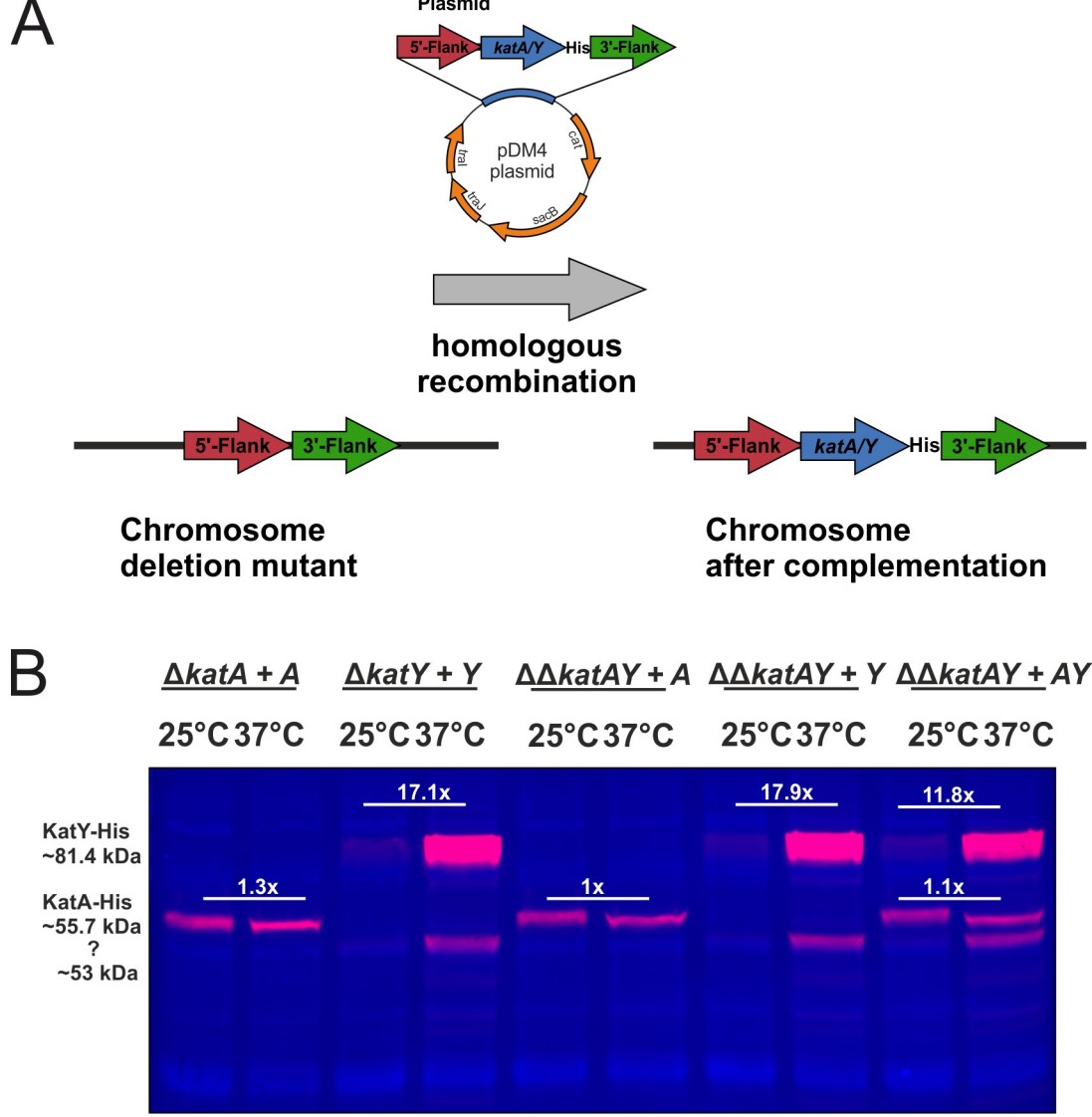

**Fig 11. KatY is the major temperature-regulated catalase in *Y. pseudotuberculosis*.** (A) The deletion strains were complemented by homologous recombination with C-terminal His-tagged KatA and KatY proteins. (B) Cells were grown to an $OD_{600}$ of 0.5 at 25 and 37°C. Samples were loaded on TGX-Stain-Free gels and the protein amount was quantitatively analysed by fluorescence detection. Normalization to the total protein amount was achieved by Stain-Free visualization. The overlay of the fluorescent detection with the Stain-Free visualisation of the total protein amount of one representative Western blot is shown. Fold differences represent the mean of three biological replicates.

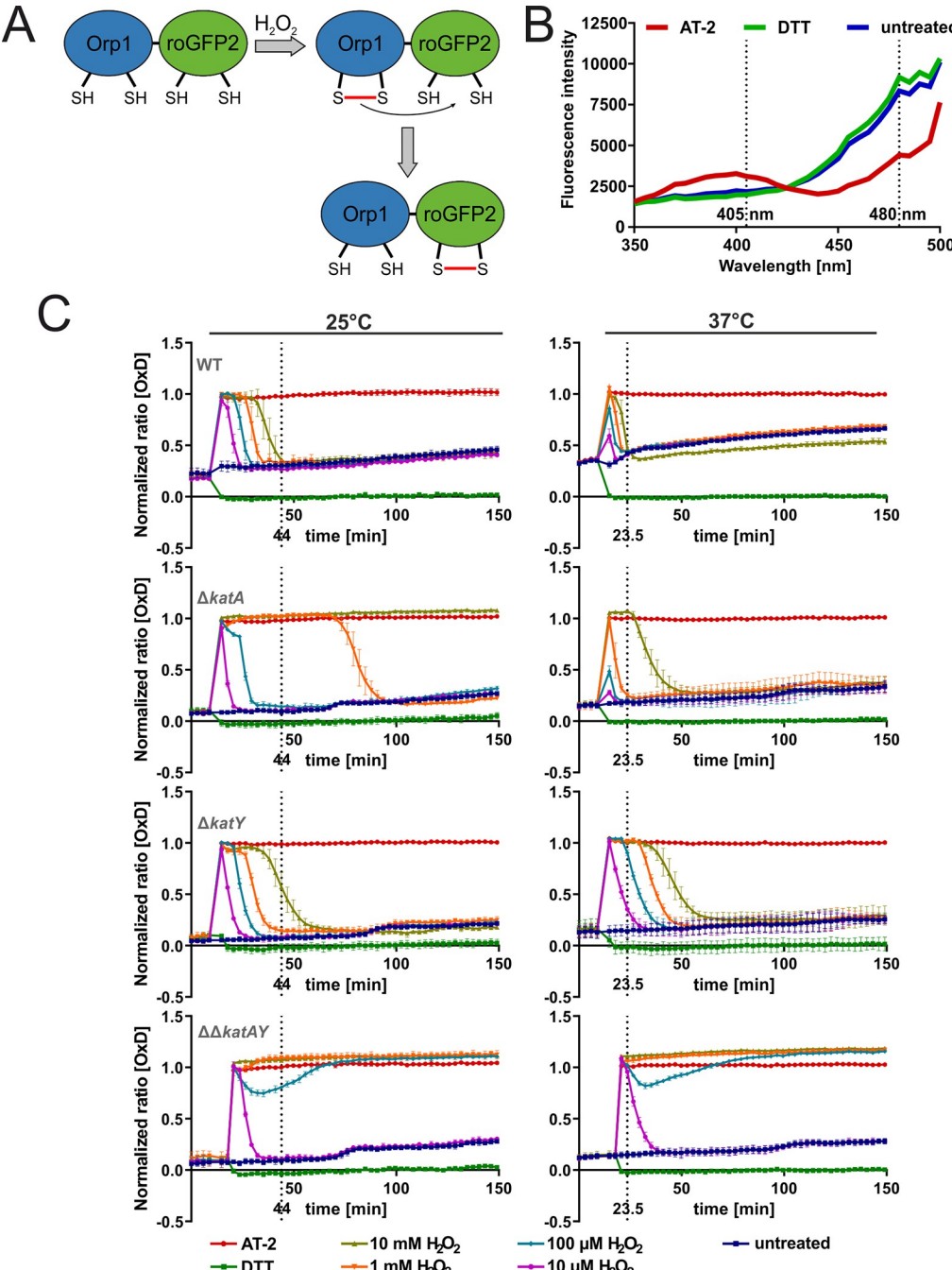

**Fig 12. KatY is responsible for the temperature dependent H₂O₂ detoxification at the intracellular level. (A)**
Schematic representation of the functionality of the roGFP2-Orp1 probe. (**B**) The roGFP2 probe exhibits different
excitation spectra based on the oxidative status. (**C**) Cells expressing the redox-sensitive roGFP2-Orp1 probes at 25˚C
or 37˚C were washed with PBS and diluted to an OD₆₀₀ of 0.2 and 100 μl were transferred to a black 96 well plate. The
fluorescence intensity at 405 nm and 488 nm was recorded over 150 min at room temperature. After measuring for 10
minutes, AT-2, DTT and H₂O₂ were added. The normalized ratio of 405/488 nm is plotted. A ratio of 1 indicates full
oxidation and a ratio of 0 indicates full reduction of the probe. The dotted line indicates the timepoint at which all H₂O₂
treated wildtype cells are returned to their untreated state. The mean and the standard deviation of biological triplicates
are plotted. Experiments were carried out multiple times.

nm. Upon thiol oxidation (e.g., by AT-2; 2,2′-Dipyridyl disulfide treatment), the intensity of this peak is reduced and another excitation peak at around 400–405 nm appears (Fig 12B). By calculating the ratio between these peaks and normalizing it to the mean ratios of fully oxidized (AT-2) and fully reduced (DTT) probes over the course of the experiment, the normalized ratio OxD is calculated [30].

*Y. pseudotuberculosis* cells harboring the roGFP2-Orp1 fusion plasmid were grown at 25 or 37˚C. Upon reaching the early exponential phase, expression of the redox probe was induced by addition of IPTG before cultures were grown at 25 or 37˚C overnight. Sufficient production and functionality of roGFP2-Orp1 was checked by measuring the excitation spectrum after addition of AT-2 and DTT. DTT, as a reductant, will fully reduce roGFP2-Orp1, while the oxidant AT-2 fully oxidizes roGFP2-Orp1 giving a readout of maximum probe reduction and oxidation before conducting the experiments. Fresh bacterial cells were then treated with AT-2, DTT or different $H_2O_2$ concentrations and directly measured for changes in their redox state at room temperature. The redox probe rapidly and fully oxidized after $H_2O_2$ application in the WT cells grown at 25˚C. Over time, the signals returned to the untreated level, reaching it after 44 min at the highest concentration of $H_2O_2$ (Fig 12C). In contrast, in the WT grown at 37˚C, the probe already returned to the untreated level at 23.5 min. Furthermore, lower $H_2O_2$ concentrations were not sufficient to fully oxidize the probe in cells grown at 37˚C, suggesting an increased protection of the intracellular space and a faster removal of $H_2O_2$ when compared to 25˚C. Consistent with the phenotypic characterization described above, the Δ*katA* mutant showed higher susceptibility against $H_2O_2$. The Δ*katA* cells grown at 25˚C were no longer able to reduce roGFP2-Orp1 after addition of 10 mM $H_2O_2$. At 37˚C, better protection against $H_2O_2$ was observed. Here, low $H_2O_2$ concentrations did not lead to full oxidation of roGFP2-Orp1, and eventually the probe is re-reduced to the untreated level with all $H_2O_2$ concentrations. The Δ*katY* strain showed a response similar to the WT at 25˚C, but interestingly unlike the WT, no increase in the cells capability to detoxify $H_2O_2$ was observed in this mutant. Since the double deletion strain ΔΔ*katAY* lacks both catalases, it shows the lowest capacity to detoxify $H_2O_2$ and to re-reduce the probe. Based on our observations that the Δ*katA* strain improves its $H_2O_2$ resistance at 37˚C and the Δ*katY* strain, while resilient as the WT at 25˚C, is not increasing its detoxification capacity at 37˚C. We conclude that the increased protection against $H_2O_2$ at 37˚C originates from the temperature induced KatY protein.

## 4. Discussion

Enteric pathogens, like *Y. pseudotuberculosis*, need to quickly sense and adapt to their changing environment upon infection. A sudden temperature upshift is among the most stable cues indicating entry into a warm-blooded host. There are many ways how bacteria can sense and respond to temperature changes. Transcription can be modulated by temperature-dependent changes in the DNA topology increasing the binding affinity of RNA polymerase to the promoter [34] or by temperature-dependent DNA-binding proteins such as the Histone-like nucleoid structuring protein H-NS [35]. In *Y. pseudotuberculosis*, H-NS forms a complex with the small YmoA protein and silences expression of *lcrF* coding for the virulence master regulator outside the host [36,37]. Translation initiation, i.e. access to the RBS, can be controlled by RNATs. [38,39]. Over the last decades, multiple RNATs have been described in various bacterial pathogens [11,40–44].

In this study, we followed up on previous reports showing that transcription and translation of several *Yersinia* ROS detoxification genes are induced at host body temperature [13–15]. Since the generation of reactive oxygen and nitrogen species constitutes the first line of host defense against microbial pathogens [45], a direct correlation between a temperature of 37˚C

and ROS defense gene expression in *Yersinia* is highly plausible. Here we aimed at understanding the underlying mechanistic details of this correlation for several classes of ROS detoxification genes.

Antioxidants, like thioredoxins are important to maintain a reduced intracellular state of proteins and remove the formed disulfide bonds, after ROS exposure [6]. In their reduced form, they can catalyze the reduction of protein disulfides and get oxidized themselves in the process. Subsequent reduction is facilitated by the thioredoxin reductase TrxB, with help of reducing equivalents from NADPH (Fig 1) [46,47]. While there is no evidence for temperature regulated transcription of the *trxA* gene, we found clear evidence for translational control of the shorter of two *trxA* transcripts. The PARS calculated RNA structures [14] show a typical RNAT-like hairpin structure with internal bulges and loops in the short 5'-UTR (Fig 4A). Stabilizing the SD/anti-SD interaction by exchange of just one residue impaired translational control, a clear indication that melting of the WT structure facilitates translation initiation, which was supported by *in vitro* experiments. The 5'-UTR of the long *trxA* adopts an entirely different structure that is unable to repress translation at 25˚C presumably accounting for sufficient TrxA levels outside of the host. The reverse is true for the *Pseudomonas aeruginosa* quorum sensing gene *pqsA* gene. Here, the longer of two alternative transcripts contains an extended 5'-end capable of folding into an RNA structure that blocks translation initiation [48]. We assume that the two different *trxA* start sites contribute to balancing TrxA levels according to the environmental situation. Previous transcriptome studies showed that the short *trxA* transcript is higher expressed than the longer *trxA* transcript (S1 Fig) [13,14]. Therefore, a high level of TrxA is guaranteed at 37˚C due to the RNAT functionality of the 5'-UTR of the short *trxA* transcript.

Superoxide dismutases (SODs) are another important defense system when bacteria face ROS challenges. *Y. pseudotuberculosis* encodes three types of SODs, an iron-containing SodB [49], a manganese-containing SodA [49,50] and a copper-containing SodC, which plays a role in *Yersinia* virulence in the *Galleria* model [51]. We expected SodB and SodC to be induced at 37˚C due to moderately increased mRNA amounts and potential RNATs in their 5'-UTRs (Figs 2, 3 and 6). Consistent with the previous PARS analysis [14], a global *in vitro* RNA structure probing approach, our biochemical data supported the existence of these RNATs (Fig 7). Reporter gene fusions, however, showed some temperature regulation in *E. coli* but essentially no repression at 25˚C in *Y. pseudotuberculosis*. Here, it is conceivable that additional post-transcriptional mechanisms come into play to coordinate appropriate *sod* gene expression. One known example in *E. coli* is the interaction of the small regulatory RNA (sRNA) RyhB, a key player in iron homeostasis [52]. RyhB negatively affects *sodB* expression by binding to the RBS of *sodB*, thereby blocking translation and facilitating cleavage by RNase E and RNase III [53,54]. RyhB expression is controlled by the ferric uptake repressor protein Fur [55], which negatively regulates RyhB expression in the presence of iron. Since iron elicits oxidative stress due to the Fenton reaction [56], an sRNA such as RyhB might perhaps help reduce the availability of iron-sulfur cluster proteins, like SodB, inside the microbial pathogen at 37˚C. In *Y. pestis* and *Y. pseudotuberculosis* the involvement of iron availability was investigated. When grown in human plasma, which represents iron-limiting conditions and compared to growth in LB media, an upregulation of *sodA* and a downregulation of *sodB* was observed [49,57]. Another report adds to the relevance of metal ions in the protection against ROS induced damage. *Y. pseudotuberculosis* uses an unconventional way to combat ROS by importing zinc as antioxidant by secretion of a zinc-binding protein via the type VI secretion system (T6SS) [58,59].

Apart from currently unexplored sRNAs in the *Yersinia* oxidative stress response, even slightly different intracellular milieus in *E. coli* and *Y. pseudotuberculosis* or the fairly dilute

conditions in the test tube might explain the observed differences between different experimental setups. It is well known that inorganic and organic cations or molecular crowding effects in the dense cellular environment can have profound effects on the folding and dynamic behavior of RNA structures [60–62]. All together we have no evidence that the RNAT candidates upstream of *sodB* and *sodC* play a significant role in controlling ROS detoxification in their native organism *Y. pseudotuberculosis*.

Finally, and most importantly, we examined the temperature regulation of *Yersinia* catalases and discovered a massive upshift of KatY at 37˚C primarily due to translational control by a novel RNAT. Even when *katY* transcription was enforced at 25 or 37˚C from a foreign promoter, the KatY protein was almost exclusively produced at 37˚C by virtue of its RNAT (S5 Fig). Catalases protect organisms against $H_2O_2$ toxicity by catalyzing its conversion into water and oxygen. KatA (also called KatE) and KatY (also called KatG) are distinct $H_2O_2$ scavenging systems. In *Y. pestis*, KatA is a monofunctional catalase functioning as primary scavenger for high levels of $H_2O_2$ whereas the bifunctional catalase/peroxidase KatY protein only shows marginal catalase activity [21]. KatY of *Y. pseudotuberculosis* is almost identical to its close relative in *Y. pestis* (99.86%). The *Y. pestis* KatY protein exists in multiple forms, most prominently an α-KatY form with a molecular mass of 78.8 kDa after processing of its leader signal sequence and a shorter β-KatY form with a molecular mass of 53.6 kDa resulting from a secondary translational start signal [22,23]. An equivalent secondary translation start site and similar translation products were observed in *Y. pseudotuberculosis* (Fig 11) supporting the close relationship between these two species.

Given this kinship, our discovery of a very efficient RNAT might explain the previously observed induction of KatY at 37˚C in *Y. pestis* [22,23], in particular since the RNAT sequence is entirely conserved between both species (S6 Fig). In contrast to a postulated ROSE (Repression Of heat Shock gene Expression)-like RNAT upstream of *katY* [21], we found a structurally unique regulatory element that reaches almost 25 nucleotides into the coding region and is one of the best acting translational silencers in *Y. pseudotuberculosis* identified to date. A combination of three strong G-C base pairs masking the SD sequence flanked by large loops might responsible for the temperature responsiveness, which can be eliminated by a single nucleotide exchange (Fig 9).

Concerning the substantial transcriptional upregulation of *katY* (Fig 2), it is interesting that potential LcrF binding sites have been postulated upstream of *katY* in *Y. pestis* [22,23]. LcrF is the master virulence regulator in *Y. pestis* and *Y. pseudotuberculosis* and itself is subject to stringent temperature control [11,40]. Having KatY production under tight dual temperature control is reminiscent of the situation of the T3SS genes *yopN*, *yscT* and *yscJ*, which are under transcriptional control by LcrF and contain their own RNATs as translational control elements [17,18].

The massive induction of KatY at host body temperature as well as its potential association with the membrane or localization in the periplasm [63,64] might suggest a critical role in virulence. At least in *Y. pestis*, however, this is not the case because a *katY* mutant was fully virulent to mice [21] suggesting a certain redundancy in the ROS protection systems. In enterohemorrhagic *E. coli* in contrast, it was shown that the secretion of a novel catalase KatN by the T6SS facilitates survival of the pathogen inside macrophages and a deletion of *katN* attenuated virulence [65].

Quite obviously, temperature is by far not the only and presumably not the dominant cue inducing the oxidative stress response in bacteria. The paradigmatic system in *E. coli* is the $H_2O_2$-activated transcription factor OxyR [66,67]. Just recently, the influence of $H_2O_2$ on the expression of ROS detoxification genes in *Y. pseudotuberculosis* was investigated [20]. A total of 364 genes were differentially regulated upon $H_2O_2$ treatment, OxyR was found to be the

master transcriptional regulator mediating cellular responses to $H_2O_2$. Among the upregulated genes were the expected ones responsible for ROS detoxification, like *trxB*, *trxC*, *katA*, *katY* and *ahpC*. The *sodB* gene coding for the iron-containing SOD enzyme was downregulated on the transcriptional level while the other SODs remained unaffected. The authors also investigated the role of KatA/KatE and KatY/KatG in $H_2O_2$ protection and detoxification at 26˚C. Consistent with our findings (Fig 10), KatA was found to be the primary catalase in *Y. pseudotuberculosis* while KatY, together with AhpR played a strong scavenging activity toward low concentrations of $H_2O_2$ [20]. Our investigation at two different temperatures added a new role of KatY in ROS protection at 37˚C. Massive induction of this catalase at mammalian body temperature might prime the pathogen for the anticipated ROS exposure in the host.

## Supporting information

**S1 Fig. The short *trxA* transcript is the more abundant isoform.**
(TIF)

**S2 Fig. Deletion of *katA* and *katY* affect susceptibility against $H_2O_2$.**
(TIF)

**S3 Fig. Temperature-dependent influence of *katA* and *katY* on the oxidative stress response is restored by complementation in *Y. pseudotuberculosis*.**
(TIF)

**S4 Fig. Temperature-dependent influence of *katA* and *katY* on the oxidative stress response during growth is restored by complementation in *Y. pseudotuberculosis*.**
(TIF)

**S5 Fig. The *katY* RNAT controls translation in a plasmid-based complementation.**
(TIF)

**S6 Fig. Sequence alignment of the upstream region of *katY* between *Y. pestis* and *Y. pseudotuberculosis*.**
(TIF)

**S1 Table. Bacterial strains.**
(DOCX)

**S2 Table. Plasmid list.**
(DOCX)

**S3 Table. Oligonucleotide list.**
(DOCX)

**S1 Data. Transcriptomic data.**
(XLSX)

**S2 Data. Beta galactosidase *bgaB* data.**
(XLSX)

**S3 Data. Zone of inhibition data.**
(XLSX)

**S4 Data. Growth curve data.**
(XLSX)

**S5 Data. roGFP2-Orp1 data.**
(XLSX)

## Acknowledgments

We thank RUBion (Central Unit for Ionbeams and Radionuclides) for the use of infrastructure. We are grateful for support with the catalase assay by Marco Krewing and Julia Bandow and thank the RNA group for continuous discussions and reading earlier versions of the manuscript.

## Author Contributions

**Conceptualization:** Daniel Scheller, Christian Twittenhoff, Franz Narberhaus.

**Formal analysis:** Daniel Scheller, Stephan Pienkoß, Christian Twittenhoff, Lisa R. Knoke.

**Funding acquisition:** Lars I. Leichert, Franz Narberhaus.

**Investigation:** Daniel Scheller, Franziska Becker, Andrea Wimbert, Dominik Meggers, Stephan Pienkoß.

**Methodology:** Daniel Scheller, Lisa R. Knoke, Lars I. Leichert.

**Project administration:** Franz Narberhaus.

**Resources:** Lars I. Leichert, Franz Narberhaus.

**Supervision:** Franz Narberhaus.

**Validation:** Daniel Scheller, Stephan Pienkoß, Lisa R. Knoke, Franz Narberhaus.

**Visualization:** Daniel Scheller.

**Writing – original draft:** Daniel Scheller.

**Writing – review & editing:** Stephan Pienkoß, Lisa R. Knoke, Lars I. Leichert, Franz Narberhaus.

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
