## [Decision Letter · Decision Letter 0]

12 Apr 2023

Dear Dr Narberhaus,

Thank you very much for submitting your Research Article entitled 'The oxidative stress response, in particular the katY gene, is temperature-regulated in Yersinia pseudotuberculosis' to PLOS Genetics.

The manuscript was fully evaluated at the editorial level and by independent peer reviewers. The reviewers appreciated the attention to an important problem, but raised some substantial concerns about the current manuscript. Based on the reviews, we will not be able to accept this version of the manuscript, but we would be willing to review a much-revised version. We cannot, of course, promise publication at that time.

Should you decide to revise the manuscript for further consideration here, your revisions should address the specific points made by reviewer 1 and reviewer 3. We will also require a detailed list of your responses to the review comments and a description of the changes you have made in the manuscript.

If you decide to revise the manuscript for further consideration at PLOS Genetics, please aim to resubmit within the next 60 days, unless it will take extra time to address the concerns of the reviewers, in which case we would appreciate an expected resubmission date by email to plosgenetics@plos.org.

We are sorry that we cannot be more positive about your manuscript at this stage. Please do not hesitate to contact us if you have any concerns or questions.

Yours sincerely,

Jue D. Wang

Academic Editor

PLOS Genetics

Lotte Søgaard-Andersen

Section Editor

PLOS Genetics

Reviewer's Responses to Questions

**Comments to the Authors:**

Reviewer #1: The oxidative stress response, in particular the katY gene, is temperature-regulated in Yersinia pseudotuberculosis

Daniel Scheller, Franziska Becker, Andrea Wimbert, Dominik Meggers, Stephan Pienkoß, Christian Twittenhoff, Lisa R. Knoke, Lars I. Leichert, Franz Narberhaus

This study investigates the relationship between RNA-thermometer (RNAT) in the context of infection and resistance to oxidative stress. These RNAT permit temperature-sensing and allow translation regulation accordingly. Four ROS defense transcripts (trxA, sodB, sodC, and katA) of Yersinia pseudotuberculosis were specifically studied for their response to temperature change (25’C vs 37’C). Overall, the transcript katY was shown to become strongly expressed through melting of inhibitory secondary structures when exposed to a temperature of 37’C. Moreover, the data convincingly indicate that KatY product is the primary H2O2 scavenger and that its expression is dependent on the temperature.

The manuscript is both clear and well written. The description of the temperature-dependent mechanism of mRNA translation activation followed by functional assays of H2O2 scavenging is quite convincing. I have a few specific comments in order to improve the quality of the manuscript.

Major comments:

1. Results in Figure 2 is missing some important points. How long was the incubation when cells were incubated from 25’C to 37’C? Please add a positive control (e.g yopN) in the figure.

2. Figure 3. Is the Western blot for Y pseudotuberculosis on the left reversed? There is a discrepancy between the B-galactosidase results and the Western blot signals (although the Western blot is not clearly indicated on the figure). Is this difference due to the decrease in transcript observed in Fig 3D? The only result fitting the Western blot and B-gal assays is the control yopN. All the others, Soda, SodB and SodC, are way stronger when extracted from 25’C compared to 37’C.

3. Figure 4. It would be interesting to visualize the WT and mutated (rep A24U) trxA mRNA by Northern or qRT-PCR. Although the results suggest a strong translational repression of trxA A24U, it is possible that the transcript becomes highly unstable due to the mutation. Please discuss this.

4. Please add a Northern blot that would show the difference in the expression ratio between the long and short endogenous isoforms of trxA.

5. Figure 5B. I understand the interest of using trxA Rep as a control for the toeprint experiment because of its inability to respond to temperature. However, this experiment would be better with a control WT mRNA that does not exhibit RNAT capacities like yopN for example.

6. While sodC is relatively constant is terms of activation through temperature, the Ecoli sodB was activated 2.5X whereas Yersinia sodB mRNA was activated 1.6X by temperature in Fig 3. This is quite different from Fig 6, where Ecoli sodB is activated 5.2X and Yersinia is activated 1.0X. Please discuss this difference.

7. Figures 4 and 6, as well as 5 and 7, could be fused together since they use pretty much the same approach and obtained similar results. The data could be summarized into a table so that we could compare the different genes more easily.

8. Figure 12. The fluorescence intensity should use a different color than yellow as it is relatively harder to compare with higher contrast colors.

9. I would move the first paragraph of the Discussion in the Introduction. As written, this paragraph is more part of a wider introduction of the subject instead of focusing on the most important results of the work.

Reviewer #2: In this manuscript, the authors investigated the potential role of RNA thermometer (RNAT) structures in the thermoregulation of genes involved in the bacterial response to oxidative stress in Yersinia pseudotuberculosis. They identified RNAT structures in the 5’UTR of five ROS defense genes and using biochemical approaches demonstrate that these predicted structures are temperature sensitive regulators of translation. Of these genes, the authors further show that temperature regulation of the expression of KatY appears to be primarily regulated post-translationally through melting of the RNAT structure in its 5’UTR. In vitro studies suggest that KatA is the primary catalase used by Y. pseudotuberculosis to protect against H2O2 at both 26 and 27 degrees. However, at 37 degrees, KatY expression increases resistance to ROS damage. Together, these data show that RNAT structures are likely significant contributors to the rapid induction of ROS defenses during Y. pseudotuberculosis infection of the mammalian host.

This was a very well written manuscript that clearly communicated the author’s rationale, hypotheses, experimental design, and results. All of the experiments included appropriate controls, and the interpretations of the results were justified by their data and previous reports in the literature. The findings significantly improve our understanding of the role of thermosensing RNAT structures in the adaptation of Y. pseudotuberculosis to the mammalian host environment. I enjoyed reading this manuscript and have no significant critiques.

Reviewer #3: This manuscript characterizes an important regulatory link between temperature sensing and the oxidative stress response in the food borne pathogen Y. pseudotuberculosis. The authors dissect the multilayered regulation of the major oxidative stress response genes, highlighting a temperature-modulated translational control via RNA structural change in the 5’UTR (i.e., RNA thermometers or RNATs).

This work is a follow-up of the authors’ previous study comparing the in vitro RNA structurome of Y. pseudotuberculosis at 37˚C vs 25˚C, which suggested possible RNATs in the oxidative stress genes. The in vitro characterization of the putative RNATs, including RNA structure probing and toeprinting, are solid. However, as detailed below, the manuscript lacks strong evidence supporting their in vivo function. 1) What is the structural difference of the proposed RNATs at 25˚C vs 37˚C in Yersinia? 2) Is the RNAT-mediated translational regulation important for oxidative stress response? These questions need to be addressed before publication.

Major points:

1) Do the proposed RNATs exhibit different structures in Yersinia at 25˚C vs 37˚C? The authors need to provide evidence showing the RNA structural change in vivo, e.g., based on data from (Twittenhoff et al., 2020). It is also unclear how the PARS-derived RNA structures shown in Fig. 4, 6, 8, 9 are predicted. Are they based on their previous publication (Righetti et al., 2016)? Are these RNA structures at 37˚C or 25˚C? The authors should cite the reference and explain how the structure model is predicted in the Material and Methods.

2) The in vivo function of putative RNATs seems rather weak in Yersinia. The authors characterized the RNATs in sodB, sodC, katA, katY and trxA-short transcripts, but only katY (and slightly for trxA-s and sodC) showed a significant temperature responsive translation increase in Yersinia, compared to the negative control sodA (Fig. 3C). The authors should discuss why the katY RNAT is more temperature sensitive than the others.

For the other genes besides katY, it is unclear how much the proposed RNATs affect their endogenous translation in Yersinia. Furthermore, the authors did not show whether these genes and their temperature sensitive regulation affect the oxidative stress response of Yersinia. Without in vivo evidence, the function of these RNATs is questionable.

3) The authors use stabilizing mutations of the proposed RNATs to examine the loss of temperature responsive regulation. However, all these mutants almost completely abolish translation of the reporter, making it difficult to interpret the results. A likely better way is introducing destabilizing base changes to the stem-loop in RNATs (but without affecting the SD sequence) to test the reduced translation inhibition at 25˚C. Additionally, a complementary mutation that recovers the base-pairing is expected to restore the RNAT function and the temperature-sensitive translation control, which would support the regulatory role of RNA structure rather than sequence.

4) The temperature regulated KatAY expression affects the susceptibility of Yersinia against H2O2. However, it is unclear how much the RNAT-mediated translation regulation contributes to it, as the mRNA level of katY is already 30-fold higher at 37˚C than at 25˚C. In addition to WT and katA/Y deletion (Fig. 10, 12), comparison to the mutants with nonfunctional RNATs (e.g., those mentioned in 3)) would significantly strengthen the conclusion.

5) The overall temperature responsive induction of genes does not reflect the combination of transcriptional and translational regulation. KatY has both its mRNA level and translation level increased ~30-fold at 37˚C vs 25˚C, measured by RNA-seq and translation reporter, respectively. However, the western blot shows only 17-fold induction (Fig. 11). The quantification of KatA induction has similar issues. The authors should address these discrepancies between different methodologies.

6) Results do not agree between figures/panels. For example, western blots in Fig. 3C (sodA, sodB, sodC) and Fig. 6C (sodC in E. coli, sodB, sodC in Yersinia) do not agree with the beta-galactosidase activity for the translational fusion reporters. The beta-galactosidase activity results of the same translational reporter are different between figures (e.g., sodB reporter in E. coli in Fig. 3B vs Fig. 6C). If different controls/adjustments were used, it should be specified in the figure legend.

Additional points:

1) Fig. 2 does not show replicates or statistical comparison of the RNA-seq and qRT-PCR data. Some data bars are missing – it is unclear whether the data is not available or is zero.

2) Comparison of mRNA level is missing for translational reporters shown in Fig. 4, 6, and 8.

3) In Fig. 10A and C, the katY deletion has higher catalase activity at 37˚C than 25˚C, but the inhibition zone of H2O2 does not show a significant difference. In general, the liquid growth (in Fig. S1) seems more sensitive than the disk diffusion assay for measuring H2O2 detoxification. The authors might consider replace the disk diffusion results with the Fig. S1.

4) In Fig. S3, why is delta katA + A more sensitive to H2O2 at 37˚C than 25˚C?

5) The protein abundance of KatY is much higher than that of KatA at 37˚C (Fig. 11). Why does KatA still play a more important role in H2O2 detoxification?

6) Table S1 does not include all the data shown in Fig. 2.

7) The authors use “anti-SD sequence” for the region that base-pairs with the SD sequence. It can be confusing because anti-SD usually refers to the 3’ end of 16S ribosomal RNA. The authors may specify its definition in the text.

**Have all data underlying the figures and results presented in the manuscript been provided?**

Reviewer #1: Yes

Reviewer #2: Yes

Reviewer #3: Yes

PLOS authors have the option to publish the peer review history of their article (what does this mean?). If published, this will include your full peer review and any attached files.

Reviewer #1: **Yes: **Eric Massé

Reviewer #2: No

Reviewer #3: No

---

## [Decision Letter · Decision Letter 1]

12 Jun 2023

Dear Dr Narberhaus,

We are pleased to inform you that your manuscript entitled "The oxidative stress response, in particular the katY gene, is temperature-regulated in Yersinia pseudotuberculosis" has been editorially accepted for publication in PLOS Genetics. Congratulations!

Yours sincerely,

Jue D. Wang

Academic Editor

PLOS Genetics

Lotte Søgaard-Andersen

Section Editor

PLOS Genetics

Comments from the reviewers (if applicable):

Reviewer's Responses to Questions

**Comments to the Authors:**

Reviewer #1: This revised version of the manuscript is now ready for publication.

Reviewer #3: The authors have provided a nicely detailed and thorough response to my comments and have addressed all my major concerns. I enjoyed reading the revised manuscript and have no further comments.

**Have all data underlying the figures and results presented in the manuscript been provided?**

Reviewer #1: Yes

Reviewer #3: None

PLOS authors have the option to publish the peer review history of their article (what does this mean?). If published, this will include your full peer review and any attached files.

Reviewer #1: No

Reviewer #3: No

**Data Deposition**

http://datadryad.org/submit?journalID=pgenetics&manu=PGENETICS-D-23-00174R1

**Press Queries**

---

## [Editor Report · Acceptance letter]

5 Jul 2023

PGENETICS-D-23-00174R1 

The oxidative stress response, in particular the katY gene, is temperature-regulated in Yersinia pseudotuberculosis 

Dear Dr Narberhaus, 

We are pleased to inform you that your manuscript entitled "The oxidative stress response, in particular the katY gene, is temperature-regulated in Yersinia pseudotuberculosis" has been formally accepted for publication in PLOS Genetics! Your manuscript is now with our production department and you will be notified of the publication date in due course.

With kind regards,

Judit Kozma

PLOS Genetics

On behalf of:
